# Associations of breastfeeding history with metabolic syndrome and cardiovascular risk factors in community-dwelling parous women: The Japan Multi-Institutional Collaborative Cohort Study

Takashi Matsunaga[1]⊙*, Yuka Kadomatsu[1‡], Mineko Tsukamoto[1‡], Yoko Kubo[1‡], Rieko Okada[1‡], Mako Nagayoshi[1‡], Takashi Tamura[1‡], Asahi Hishida[1‡], Toshiro Takezaki[2‡], Ippei Shimoshikiryo[2‡], Sadao Suzuki[3‡], Hiroko Nakagawa[3‡], Naoyuki Takashima[4,5‡], Yoshino Saito[4,6‡], Kiyonori Kuriki[7‡], Kokichi Arisawa[8‡], Sakurako Katsuura-Kamano[8‡], Nagato Kuriyama[9‡], Daisuke Matsui[9‡], Haruo Mikami[10‡], Yohko Nakamura[10‡], Isao Oze[11‡], Hidemi Ito[12,13‡], Masayuki Murata[14‡], Hiroaki Ikezaki[14‡], Yuichiro Nishida[15‡], Chisato Shimanoe[16‡], Kenji Takeuchi[1‡], Kenji Wakai[1]⊙

1 Department of Preventive Medicine, Nagoya University Graduate School of Medicine, Nagoya, Aichi, Japan, 2 Department of International Island and Community Medicine, Kagoshima University Graduate School of Medical and Dental Sciences, Kagoshima, Kagoshima, Japan, 3 Department of Public Health, Nagoya City University Graduate School of Medical Sciences, Nagoya, Aichi, Japan, 4 Department of Public Health, Shiga University of Medical Science, Otsu, Shiga, Japan, 5 Department of Public Health, Faculty of Medicine, Kindai University, Osaka-Sayama, Osaka, Japan, 6 Department of Nursing, Faculty of Health Science, Aino University, Ibaraki, Osaka, Japan, 7 Laboratory of Public Health, Division of Nutritional Sciences, School of Food and Nutritional Sciences, University of Shizuoka, Shizuoka, Shizuoka, Japan, 8 Department of Preventive Medicine, Tokushima University Graduate School of Biomedical Sciences, Tokushima, Tokushima, Japan, 9 Department of Epidemiology for Community Health and Medicine, Kyoto Prefectural University of Medicine, Kyoto, Kyoto, Japan, 10 Cancer Prevention Center, Chiba Cancer Center Research Institute, Chiba, Chiba, Japan, 11 Division of Cancer Epidemiology and Prevention, Aichi Cancer Center Research Institute, Nagoya, Aichi, Japan, 12 Division of Cancer Information and Control, Aichi Cancer Center Research Institute, Nagoya, Aichi, Japan, 13 Division of Descriptive Cancer Epidemiology, Nagoya University Graduate School of Medicine, Nagoya, Aichi, Japan, 14 Department of General Internal Medicine, Kyushu University Hospital, Fukuoka, Fukuoka, Japan, 15 Department of Preventive Medicine, Faculty of Medicine, Saga University, Saga, Saga, Japan, 16 Department of Pharmacy, Saga University Hospital, Saga, Saga, Japan

⊙ These authors contributed equally to this work.
‡ YK, MT, YK, RO, MN, TT, AH, TT, IS, SS, HN, NT, YS, KK, KA, SK, NK, DM, HM, YN, IO, HI, MM, HI, YN, CS, and KT also contributed equally to this work.
* matsunaga.takashi@d.mbox.nagoya-u.ac.jp

**Data Availability Statement:** The data used in the present study cannot be made publicly available

## Abstract

### Objective

The aim of the present study was to investigate the associations between breastfeeding and the prevalence of metabolic syndrome in community-dwelling parous women and to clarify whether the associations depend on age.

### Methods

The present cross-sectional study included 11,118 women, aged 35–69 years. Participants' longest breastfeeding duration for one child and their number of breastfed children were

because the participants of the present study have not given informed consent for public data sharing and the ethics committee of the Nagoya University Graduate School of Medicine does not approve the sharing. To obtain access to the confidential data, interested researchers can contact the Administration Office of the ethics committee of the Nagoya University Graduate School of Medicine (iga-shinsa@adm.nagoya-u.ac.jp). After receiving a request, informed consent from the participants will be sought.

**Funding:** This study was supported by Grants-in-Aid for Scientific Research for Priority Areas of Cancer (No. 17015018) and Innovative Areas (No. 221S0001) from the Japan Society for the Promotion of Science, and by the Platform of Supporting Cohort Study and Biospecimen Analysis (CoBiA, JSPS KAKENHI Grant Number 16H06277) from the Japanese Ministry of Education, Culture, Sports, Science and Technology. The funders had no role in the study design, data collection and analysis, decision to publish, or preparation of the manuscript.

**Competing interests:** The authors have declared no conflict of interest in the present study.

assessed using a self-administered questionnaire, and their total breastfeeding duration was approximated as a product of the number of breastfed children and the longest breastfeeding duration. The longest and the total breastfeeding durations were categorized into none and tertiles above 0 months. Metabolic syndrome and cardiovascular risk factors (obesity, hypertension, dyslipidemia, and hyperglycemia) were defined as primary and secondary outcomes, respectively. Associations between breastfeeding history and metabolic syndrome or each cardiovascular risk factor were assessed using multivariable unconditional logistic regression analysis.

## Results

Among a total of 11,118 women, 10,432 (93.8%) had ever breastfed, and 1,236 (11.1%) had metabolic syndrome. In participants aged <55 years, an inverse dose–response relationship was found between the number of breastfed children and the prevalence of metabolic syndrome; multivariable-adjusted odds ratios for 1, 2, 3, and ≥4 breastfed children were 0.60 (95% confidence interval [CI]: 0.31 to 1.17), 0.50 (95% CI: 0.29 to 0.87), 0.44 (95% CI: 0.24 to 0.84), and 0.35 (95% CI: 0.14 to 0.89), respectively. The longest and total breastfeeding durations of longer than 0 months were also associated with lower odds of metabolic syndrome relative to no breastfeeding history in participants aged <55 years. In contrast, all measures of breastfeeding history were not significantly associated with metabolic syndrome and cardiovascular risk factors in participants aged ≥55 years old.

## Conclusions

Breastfeeding history may be related to lower prevalence of metabolic syndrome in middle-aged parous women.

## Introduction

Metabolic syndrome is characterized as a cluster of cardiovascular risk factors, including central obesity, hypertension, dyslipidemia, and glucose intolerance [1]. Recent meta-analyses have found that metabolic syndrome increases the risks of cardiovascular disease [2], cancers [3], chronic kidney disease [4], liver-related events [5], and all-cause mortality [6]. Sixty-three percent of deaths from cardiovascular diseases, chronic kidney diseases, and diabetes mellitus in the world in 2010 were estimated to be attributable to the combined effect of high blood pressure, blood glucose, serum cholesterol, and body mass index [7]. The prevalence of metabolic syndrome varies by population and definition, but it is high and has been steadily increasing in developed and developing countries alike [1]. Therefore, metabolic syndrome is imposing a huge burden on the worldwide healthcare system, and effective measures for prevention are needed.

Previous studies have identified that breastfeeding has beneficial health effects on mothers as well as children, including reduced risks of breast, ovarian, and endometrial cancers in mothers [8, 9]. Meta-analyses of observational studies have shown that breastfeeding has some protective effects on the maternal risks of type 2 diabetes mellitus [10], hypertension [11], and postpartum weight retention [12], and some observational studies have suggested its protective effects against metabolic syndrome [13–18] and cardiovascular disease [19]. As a biological

mechanism behind these effects, the reset hypothesis suggests that pregnancy induces fat accumulation and increases in insulin resistance and lipid levels in anticipation of the metabolic needs for breastfeeding, and that breastfeeding reverses these responses [20].

However, some issues need to be resolved. First, not all previous studies found inverse associations between breastfeeding and metabolic syndrome [21–24], and their results were inconsistent. Second, some studies have reported that the protective effects of breastfeeding against hypertension [25], type 2 diabetes mellitus [26], cardiovascular risk factors [27], and cardiovascular disease [28] are limited to younger individuals. However, only one study investigating its effects on metabolic syndrome has conducted stratified analyses by age [22]. Moreover, means or medians of participants' ages were the early fifties or younger (<55 years old) in most cross-sectional studies [13, 15–17, 22, 23] or at the end of follow-up among all cohort studies [14, 24], such that the effects of breastfeeding history on metabolic syndrome later in life are still unclear.

In the present study, we aimed to investigate the associations of breastfeeding with the prevalence of metabolic syndrome in community-dwelling parous women and to clarify whether the associations depend on age.

## Materials and methods

### Study design and participants

The present study was conducted using the baseline data of the Japan Multi-Institutional Collaborative Cohort (J-MICC) Study. Details of the J-MICC Study were described elsewhere [29]. The study mainly aims to examine gene-environment interactions in lifestyle-related diseases, especially cancers. Participants provided written informed consent prior to participation. The study protocol was approved by the ethics committee of Nagoya University Graduate School of Medicine (approval number: 2010-0939-7) and other institutions participating in the J-MICC Study.

The baseline surveys were conducted between February 2004 and March 2014 in 14 research areas throughout Japan. Participants aged 35–69 years at the baseline surveys were recruited from community-dwelling adults, company employees, health check examinees, and first-visit patients at a cancer hospital. Lifestyle, drug usage, and reproductive history were surveyed using a self-administered questionnaire. Serum lipids, blood glucose, blood pressure, height, and weight were measured at the health checkup. As of February 15, 2019, 92,631 participants were recruited. The dataset version 20190215 was used for the present analyses. Of the 14 research areas, five did not collect breastfeeding history and/or biochemical data, and thus 49,162 participants in these areas were excluded, leaving 43,469 persons. Of them, we further excluded 20,518 men and 3,157 women who had not delivered a baby, 290 persons with missing values for measurements of breastfeeding, 5,118 with one or more missing biochemical or drug data, 54 whose serum triglycerides were 400 mg/dL or higher, and 3,214 whose postprandial time before the blood draw was shorter than 8 hours or missing, leaving 11,118 parous women for analyses. To examine the possibility of reverse causality where body fat before delivery could prevent initiation or maintenance of breastfeeding [30], 314 participants who did not report their body weight at about 20 years of age were further excluded, thus leaving 10,804 participants for this analysis (Fig 1).

### Exposures

A self-administered questionnaire regarding breastfeeding history was administered. The number of breastfed children and the longest breastfeeding duration among all the breastfed children were assessed using two questions: "How many children have you ever breastfed in

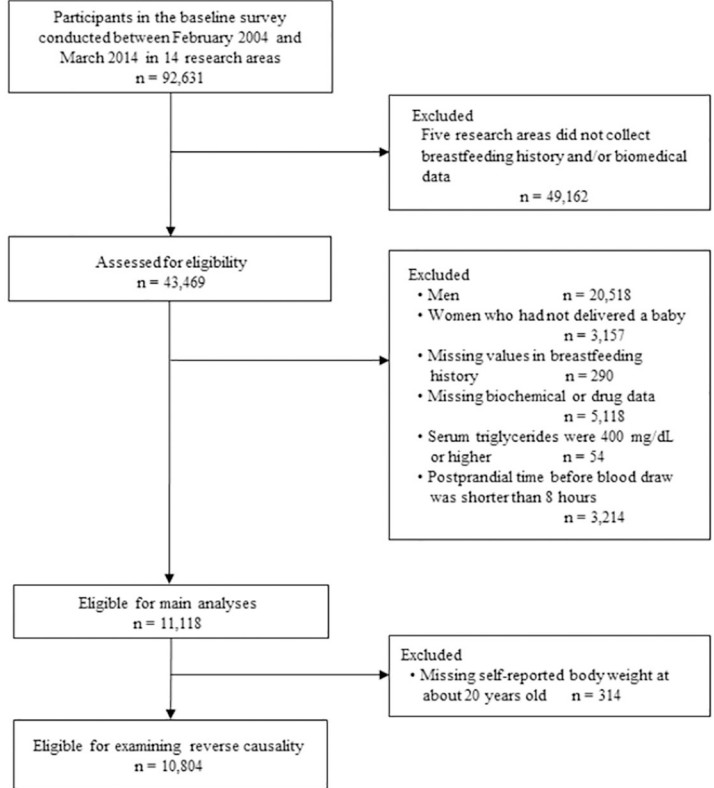

**Fig 1. Flow diagram showing selection of eligible participants.**

total?" and "How long was your longest duration of breastfeeding in years and months or weeks after childbirth?" Due to this limitation of the questionnaire, the total breastfeeding duration was approximated as a product of the number of breastfed children and the longest breastfeeding duration. Our questionnaire did not have items concerning exclusive breastfeeding (where an infant receives only breast milk without any additional food or drink including water). The number of breastfed children was categorized into 0, 1, 2, 3, and ≥4 (five groups in total), and the longest breastfeeding duration and the total breastfeeding duration were categorized into 0 and tertiles among women with breastfeeding history (four groups in total) to deal with potential nonlinear associations.

## Outcomes

Metabolic syndrome was defined as the primary outcome and cardiovascular risk factors as the secondary outcomes. The prevalence of metabolic syndrome and cardiovascular risk factors were assessed by referring to the revised National Cholesterol Education Program Adult Treatment Panel III criteria [31]. Participants were diagnosed with metabolic syndrome if they satisfied at least three of the following five criteria: (a) obesity: body mass index (BMI) ≥25 kg/m$^2$ instead of elevated waist circumference; (b) elevated blood pressure: systolic blood pressure ≥130 mm Hg, diastolic blood pressure ≥85 mm Hg, and/or self-reported use of antihypertensive drugs; (c) elevated triglycerides: serum triglycerides ≥150 mg/dL; (d) reduced high-density lipoprotein cholesterol (HDL-C): serum HDL-C <50 mg/dL; (e) elevated fasting glucose: fasting blood glucose ≥100 mg/dL and/or self-reported use of antidiabetic drugs. In addition,

the serum concentration of low-density lipoprotein cholesterol (LDL-C) was calculated using the Friedewald formula (LDL-C = total cholesterol − HDL-C − triglycerides/5). Elevated LDL-C was defined as serum LDL-C $\geq$140 mg/dL and/or use of cholesterol-lowering drugs, and dyslipidemia was defined as the presence of at least one of the following three criteria: elevated LDL-C, elevated triglycerides, and reduced HDL-C.

## Covariates

When assessing associations of each breastfeeding measurement with metabolic syndrome and cardiovascular risk factors, the following demographic, lifestyle, and reproductive factors were considered as covariates: age (continuous), residential area (Tokai, Kinki, Shikoku or Kyushu), educational attainment (elementary or junior high school, high school or junior college, university or graduate school, missing), smoking status (never, former, current, missing), alcohol consumption (for <55 years old at the baseline survey: 0.0, 0.1–6.4, $\geq$6.5 g/day, missing; for $\geq$55 years old: 0.0, 0.1–5.1, $\geq$5.2 g/day, missing; 6.5 and 5.2 g/day were the medians above 0 g/day, respectively), daily total physical activity (for <55 years old: <14.0, 14.0–26.4, $\geq$26.5 metabolic equivalent hours per day [MET-h/day], missing; for $\geq$55 years old: <16.3, 16.3–28.2, $\geq$28.3 MET-h/day, missing; 14.0 and 26.5 MET-h/day and 16.3 and 28.3 MET-h/day were the cut-offs for tertiles above 0 MET-h/day, respectively), leisure-time physical activity (for <55 years old: 0.0, 0.1–1.5, $\geq$1.6 MET-h/day, missing; for $\geq$55 years old: 0.0, 0.1–2.1, $\geq$2.2 MET-h/day, missing; 1.6 and 2.2 MET-h/day were the medians above 0 MET-h/day, respectively), the modified Japanese Diet Index (for <55 years old: <2.0, 2.0–4.9, $\geq$5.0 points, missing; for $\geq$55 years old: <3.0, 3.0–5.9, $\geq$6.0 points, missing; 2.0 and 5.0 points and 3.0 and 6.0 points were the cut-offs for tertiles, respectively) [31], parity (1, 2, 3, $\geq$4, missing), age at first delivery (for <55 years old: <25, 25–27, $\geq$28 years old, missing; for $\geq$55 years old: <24, 24–25, $\geq$26 years old, missing; 25 and 28 years old and 24 and 26 years old were the cut-offs for tertiles, respectively), and history of hormone replacement therapy (never, former or current, missing). The cut-offs of continuous variables were determined based on the distributions of the present population and generally agreed with those reported by previous Japanese studies [32–34].

Alcohol consumption was calculated as g/day ethanol based on the dose and frequency of drinking of each alcoholic beverage. Daily total physical activity and leisure-time physical activity were calculated as MET-h/day using the intensity, duration, and frequency of three levels of daily or leisure-time physical activity (daily total physical activity includes laborious activity, walking, and standing; leisure-time physical activity includes vigorous, moderate, and light activity). Quality of diet was evaluated using the modified Japanese Diet Index based on the frequency of consumption of each food; pickled vegetables were excluded from the index because of no data, and thus, this index ranged from −2 to 9 in the present study [35].

## Statistical analysis

Participants' characteristics are expressed as means ± standard deviations for continuous variables and as numbers and proportions (%) for categorical variables according to the categories of the total breastfeeding duration. Quantile-quantile plots suggested deviation from normality for all the continuous variables. Thus, associations between participant characteristics and the total breastfeeding duration were examined using the Kruskal-Wallis test for continuous variables. The chi-square test of independence was applied to categorical variables.

Associations of each measurement of breastfeeding history with metabolic syndrome and cardiovascular risk factors were examined using unconditional logistic regression models, and odds ratios (ORs) for the outcomes were estimated with their 95% confidence intervals (CIs).

Two models were established: 1) age-adjusted model (model 1) and 2) multivariable-adjusted model including all the covariates listed in section "4. Covariates" (model 2). All multivariable-adjusted regression models incorporated missing categories for covariates with missing values (educational attainment, smoking status, alcohol consumption, daily total physical activity, leisure-time physical activity, the modified Japanese Diet Index, parity, age at first delivery, and history of hormone replacement therapy). A trend test was also performed by entering a variable scored as 0, 1, 2, 3, or 4 for the four or five groups with increasing breastfeeding history as a single variable in the model. Each model was built considering all participants and by age category (<55 and ≥55 years old at the baseline survey) because the means or medians of participants' age had been 55 years or younger in most of the previous studies that examined the association between breastfeeding history and metabolic syndrome [13–17, 22–24]. Likelihood ratio tests were conducted to examine the effect modification by age on the associations between breastfeeding history and metabolic syndrome. These tests calculated the *P* values of the product terms of breastfeeding history (ordinal variable) and age (continuous variable) in the aforementioned models 1 and 2.

Two sensitivity analyses were conducted. First, when examining the associations between breastfeeding history and metabolic syndrome, the longest breastfeeding duration, the number of breastfed children, and the total breastfeeding duration was incorporated as a continuous variable into models 1 and 2 rather than categorical variables or an ordinal variable. Second, when examining the same associations, we established a multivariable-adjusted model that incorporated only covariates that had preceded breastfeeding history in time (age, residential area, educational attainment, age at first delivery, and parity).

Additional analyses were conducted to examine the possibility of reverse causality where body fat before delivery prevents initiation or maintenance of breastfeeding. In these analyses, BMI was calculated using self-reported weight at about 20 years of age and height at the baseline survey and was categorized into <18.5, 18.5–22.9, and ≥23.0 kg/m$^2$ by referring to the World Health Organization's classification of BMI for Asian people [36]. Associations between BMI at about 20 years of age and each measurement of breastfeeding history were examined using unconditional binary logistic regression analysis, and ORs for each measurement of breastfeeding history were estimated with their 95% CIs because quantile-quantile plots indicated that the longest and the total breastfeeding duration deviated from normality. In these analyses, outcomes were defined as follows in all participants, those <55 years old, and those ≥55 years old: longest breastfeeding duration of ≥8.1, ≥8.1, and ≥7.1 months, respectively; number of breastfed children ≥2 children in all three groups; and total breastfeeding duration of ≥12.1, ≥15.7, and ≥12.1 months, respectively. These cut-offs for breastfeeding history were selected from the cut-offs used to analyze the associations between breastfeeding history and metabolic syndrome. We established a multivariable-adjusted model that incorporated the same covariates as the aforementioned model 2 except for the analyses that incorporated the number of breastfed children as an outcome. Because parity strongly predicted the number of breastfed children, the multivariable models incorporating the number of breastfed children as an outcome excluded parity from the covariates. Trend tests were also conducted by entering a variable scored as 0, 1, or 2 for the three groups of BMI (<18.5, 18.5–22.9, and ≥23.0 kg/m$^2$) as a single variable in the model.

All statistical tests were two-sided. The Bonferroni correction was conducted for the trend tests examining the associations between breastfeeding history and metabolic syndrome, which was the primary outcome. Because the present study included three exposure variables (the longest breastfeeding duration, number of breastfed children, and total breastfeeding duration), the significance level was set at 0.05/3 ≈ 0.017. Regarding the tests examining the association between categorized breastfeeding history and metabolic syndrome with a

reference category, $P < 0.05$ was considered statistically significant because observational epidemiological studies rarely conduct corrections for multiple comparisons on such analyses. Furthermore, in the analyses that incorporated cardiovascular risk factors as the secondary outcomes or examined the effect modification or reverse causality and in the sensitivity analyses, $P < 0.05$ was considered statistically significant. Stata 13.1 (Stata Corp. LP, College Station, TX, USA) was used for statistical analyses.

## Results

Among a total of 11,118 participants, 686 (6.2%), 3,104 (27.9%), 3,263 (29.3%), and 4,065 (36.6%) reported 0, 0.1–12.0, 12.1–30.0, and ≥30.1 months, respectively, of total breastfeeding duration (the product of the number of breastfed children and the longest breastfeeding duration, which was categorized into 0 and tertiles among women with breastfeeding history), and 1,236 had metabolic syndrome (prevalence: 11.1%). Table 1 compares characteristics of participants among the categories of total breastfeeding duration by age group. In participants aged <55 years old, those with a longer total breastfeeding duration tended to have higher educational attainment, to be a never smoker, to participate in higher daily total and leisure-time physical activity, to consume a high-quality diet, to be multiparous, to be younger at their first delivery, and to not have a history of use of hormone replacement therapy. Furthermore, in the younger age group, those with a longer total breastfeeding duration were likely to show lower systolic blood pressure, lower serum LDL-C and blood glucose, and to not take antihypertensive drugs. On the other hand, in participants ≥55 years old, those with a longer total breastfeeding duration tended to live in the Shikoku or Kyushu area, to have lower educational attainment, to be a never smoker, to participate in higher daily total and leisure-time physical activity, to consume a high-quality diet, to be multiparous and younger at their first delivery, and to not have a history of use of hormone replacement therapy. Moreover, in the older age group, those with a longer total breastfeeding duration tended to have a higher body mass index, to have a decreased level of HDL-C and an increased level of blood glucose, and to take antihypertensive drugs.

Table 2 shows the associations between metabolic syndrome and the longest breastfeeding duration, the number of breastfed children, and the total breastfeeding duration. Among all participants, we found no associations between any measurements of breastfeeding history and metabolic syndrome. However, in the analysis limited to participants aged <55 years old, the Bonferroni corrections indicated an inverse dose–response relationship between the number of breastfed children and the prevalence of metabolic syndrome; multivariable-adjusted ORs for 1, 2, 3, and ≥4 breastfed children were 0.60 (95% CI: 0.31 to 1.17), 0.50 (95% CI: 0.29 to 0.87), 0.44 (95% CI: 0.24 to 0.84), and 0.35 (95% CI: 0.14 to 0.89), respectively ($P$ trend = 0.006). In this younger group, the longest and total breastfeeding durations of longer than 0 months were also associated with lower odds of metabolic syndrome relative to no breastfeeding history; multivariable-adjusted ORs for 0.1–8.0, 8.1–12.0, and ≥12.1 months of the longest breastfeeding duration were 0.49 (95% CI: 0.29 to 0.83), 0.58 (95% CI: 0.35 to 0.97), and 0.45 (95% CI: 0.27 to 0.78), respectively ($P$ trend = 0.123); multivariable-adjusted ORs for 0.1–15.6, 15.7–32.0, and ≥32.1 months of total breastfeeding duration were 0.49 (95% CI: 0.29 to 0.82), 0.58 (95% CI: 0.35 to 0.98), and 0.48 (95% CI: 0.28 to 0.82), respectively ($P$ trend = 0.175). On the other hand, in participants aged ≥55 years old, we found no associations of metabolic syndrome with the longest breastfeeding duration, the number of breastfed children, or the total breastfeeding duration. In likelihood ratio tests, $P$ values for interaction of the longest breastfeeding duration, number of breastfed children, and total breastfeeding duration with age were <0.001, 0.005, and 0.002, respectively, in the age-adjusted model, and

**Table 1. Characteristics of participants according to total breastfeeding duration by age group.**

<55 years old

| Characteristics | Total breastfeeding duration (months)[a] | | | | |
|---|---|---|---|---|---|
| | 0 | 0.1–15.6 | 15.7–32.0 | ≥32.1 | P value[b] |
| Number of participants | 164 | 1,395 | 1,534 | 1,580 | |
| Age (years), mean ± SD | 47.0 ± 5.1 | 46.4 ± 5.4 | 46.2 ± 5.4 | 46.6 ± 5.3 | 0.186[c] |
| Residential area, n (%) | | | | | |
| Tokai area | 89 (54.3) | 855 (61.3) | 948 (61.8) | 846 (53.5) | <0.001[d] |
| Kinki area | 21 (12.8) | 198 (14.2) | 205 (13.4) | 201 (12.7) | |
| Shikoku or Kyushu area | 54 (32.9) | 342 (24.5) | 381 (24.8) | 533 (33.7) | |
| Educational attainment, n (%) | | | | | |
| Elementary or junior high school | 86 (52.4) | 621 (44.5) | 597 (38.9) | 709 (44.9) | <0.001[d] |
| High school or junior college | 43 (26.2) | 512 (36.7) | 612 (39.9) | 626 (39.6) | |
| University or graduate school | 12 (7.3) | 132 (9.5) | 213 (13.9) | 174 (11.0) | |
| Missing | 23 (14.0) | 130 (9.3) | 112 (7.3) | 71 (4.5) | |
| Smoking status, n (%) | | | | | |
| Never | 123 (75.0) | 1,077 (77.2) | 1,324 (86.3) | 1,368 (86.6) | <0.001[d] |
| Former | 20 (12.2) | 133 (9.5) | 99 (6.5) | 106 (6.7) | |
| Current | 20 (12.2) | 159 (11.4) | 93 (6.1) | 93 (5.9) | |
| Missing | 1 (0.6) | 26 (1.9) | 18 (1.2) | 13 (0.8) | |
| Alcohol consumption (g/day), n (%) | | | | | |
| 0.0 | 106 (64.6) | 866 (62.1) | 916 (59.7) | 948 (60.0) | 0.134[d] |
| 0.1–6.4 | 23 (14.0) | 245 (17.6) | 325 (21.2) | 307 (19.4) | |
| ≥6.5 | 35 (21.3) | 284 (20.4) | 292 (19.0) | 324 (20.5) | |
| Missing | 0 (0.0) | 0 (0.0) | 1 (0.1) | 1 (0.1) | |
| Daily total physical activity (MET-h/day), n (%) | | | | | |
| <14.0 | 67 (40.9) | 466 (33.4) | 503 (32.8) | 449 (28.4) | 0.004[d] |
| 14.0–26.4 | 53 (32.3) | 470 (33.7) | 536 (34.9) | 560 (35.4) | |
| ≥26.5 | 43 (26.2) | 459 (32.9) | 494 (32.2) | 571 (36.1) | |
| Missing | 1 (0.6) | 0 (0.0) | 1 (0.1) | 0 (0.0) | |
| Leisure-time physical activity (MET-h/day), n (%) | | | | | |
| 0.0 | 65 (39.6) | 429 (30.8) | 441 (28.8) | 509 (32.2) | <0.001[d] |
| 0.1–1.5 | 69 (42.1) | 645 (46.2) | 677 (44.1) | 618 (39.1) | |
| ≥1.6 | 30 (18.3) | 321 (23.0) | 416 (27.1) | 453 (28.7) | |
| The modified Japanese Diet Index (points)[e], n (%) | | | | | |
| <2.0 | 54 (32.9) | 427 (30.6) | 422 (27.5) | 462 (29.2) | 0.046[d] |
| 2.0–4.9 | 53 (32.3) | 406 (29.1) | 439 (28.6) | 415 (26.3) | |
| ≥5.0 | 57 (34.8) | 562 (40.3) | 673 (43.9) | 703 (44.5) | |
| Parity, n (%) | | | | | |
| 1 | 48 (29.3) | 402 (28.8) | 92 (6.0) | 32 (2.0) | <0.001[d] |
| 2 | 76 (46.3) | 733 (52.5) | 992 (64.7) | 403 (25.5) | |
| 3 | 31 (18.9) | 219 (15.7) | 373 (24.3) | 835 (52.9) | |
| ≥4 | 5 (3.1) | 32 (2.3) | 69 (4.5) | 305 (19.3) | |
| Missing | 4 (2.4) | 9 (0.7) | 8 (0.5) | 5 (0.3) | |
| Age at first delivery (years), n (%) | | | | | |
| <25 | 52 (31.7) | 436 (31.3) | 462 (30.1) | 586 (37.1) | <0.001[d] |
| 25–27 | 47 (28.7) | 392 (28.1) | 530 (34.6) | 537 (34.0) | |
| ≥28 | 58 (35.4) | 550 (39.4) | 529 (34.5) | 443 (28.0) | |
| Missing | 7 (4.3) | 17 (1.2) | 13 (0.9) | 14 (0.9) | |

(*Continued*)

**Table 1.** (Continued)

| | | | | | |
|---|---|---|---|---|---|
| History of hormone replacement therapy, n (%) | | | | | |
| Never | 142 (86.6) | 1,203 (86.2) | 1,375 (89.6) | 1,433 (90.7) | 0.001[d] |
| Former or current | 20 (12.2) | 186 (13.3) | 155 (10.1) | 141 (8.9) | |
| Missing | 2 (1.2) | 6 (0.4) | 4 (0.3) | 6 (0.4) | |
| Body mass index (kg/m$^2$), mean ± SD | 23.0 ± 4.3 | 22.1 ± 3.3 | 22.2 ± 3.4 | 22.5 ± 3.4 | <0.001[c] |
| Systolic blood pressure (mm Hg), mean ± SD | 120.6 ± 17.8 | 117.9 ± 17.2 | 116.8 ± 16.7 | 117.3 ± 17.1 | 0.032[c] |
| Diastolic blood pressure (mm Hg), mean ± SD | 74.7 ± 11.3 | 72.9 ± 11.2 | 72.4 ± 10.9 | 72.8 ± 10.8 | 0.104[c] |
| Use of antihypertensive drugs, n (%) | | | | | |
| No | 147 (89.6) | 1,298 (93.1) | 1,451 (94.6) | 1,463 (92.6) | 0.030[d] |
| Yes | 17 (10.4) | 97 (7.0) | 83 (5.4) | 117 (7.4) | |
| Serum LDL-C (mg/dL), mean ± SD | 121.1 ± 29.7 | 119.4± 30.8 | 116.0 ± 30.2 | 116.7 ± 30.5 | 0.002[c] |
| Serum triglycerides (mg/dL), mean ± SD | 86.3 ± 46.8 | 82.0 ± 42.9 | 79.3 ± 42.2 | 81.7 ± 44.1 | 0.081[c] |
| Serum HDL-C (mg/dL), mean ± SD | 71.1 ± 17.3 | 70.7 ± 16.4 | 71.7 ± 16.4 | 70.0 ± 15.8 | 0.014[c] |
| Use of cholesterol-lowering drugs, n (%) | | | | | |
| No | 156 (95.1) | 1,343 (96.3) | 1,474 (96.1) | 1,536 (97.2) | 0.232[d] |
| Yes | 8 (4.9) | 52 (3.7) | 60 (3.9) | 44 (2.8) | |
| Blood glucose (mg/dL), mean ± SD | 93.2 ± 11.1 | 91.3 ± 11.2 | 91.5 ± 12.2 | 91.4 ± 12.0 | 0.039[c] |
| Use of antidiabetic drugs, n (%) | | | | | |
| No | 163 (99.4) | 1,385 (99.3) | 1,519 (99.0) | 1,565 (99.1) | 0.845[d] |
| Yes | 1 (0.6) | 10 (0.7) | 15 (1.0) | 15 (1.0) | |

≥55 years old

| Characteristics | Total breastfeeding duration (months)[a] | | | | |
|---|---|---|---|---|---|
| | 0 | 0.1–12.0 | 12.1–30.0 | ≥30.1 | P value[b] |
| Number of participants | 522 | 1,663 | 1,961 | 2,299 | |
| Age (years), mean ± SD | 62.3 ± 4.1 | 61.6 ± 4.0 | 61.8 ± 4.2 | 62.2 ± 4.2 | <0.001[c] |
| Residential area, n (%) | | | | | |
| Tokai area | 236 (45.2) | 942 (56.6) | 911 (46.5) | 829 (36.1) | <0.001[d] |
| Kinki area | 97 (18.6) | 287 (17.3) | 273 (13.9) | 325 (14.1) | |
| Shikoku or Kyushu area | 189 (36.2) | 434 (26.1) | 777 (39.6) | 1,145 (49.8) | |
| Educational attainment, n (%) | | | | | |
| Elementary or junior high school | 379 (72.6) | 1,106 (66.5) | 1,394 (71.1) | 1,776 (77.3) | <0.001[d] |
| High school or junior college | 97 (18.6) | 367 (22.1) | 404 (20.6) | 395 (17.2) | |
| University or graduate school | 20 (3.8) | 112 (6.7) | 97 (5.0) | 80 (3.5) | |
| Missing | 26 (5.0) | 78 (4.7) | 66 (3.4) | 48 (2.1) | |
| Smoking status, n (%) | | | | | |
| Never | 461 (88.3) | 1,541 (92.7) | 1,820 (92.8) | 2,186 (95.1) | <0.001[d] |
| Former | 28 (5.4) | 55 (3.3) | 75 (3.8) | 59 (2.6) | |
| Current | 24 (4.6) | 60 (3.6) | 51 (2.6) | 46 (2.0) | |
| Missing | 9 (1.7) | 7 (0.4) | 15 (0.8) | 8 (0.4) | |
| Alcohol consumption (g/day), n (%) | | | | | |
| 0.0 | 395 (75.7) | 1,233 (74.1) | 1,463 (74.6) | 1,700 (74.0) | 0.184[d] |
| 0.1–5.1 | 59 (11.3) | 239 (14.4) | 243 (12.4) | 285 (12.4) | |
| ≥5.2 | 68 (13.0) | 189 (11.4) | 255 (13.0) | 313 (13.6) | |
| Missing | 0 (0.0) | 2 (0.1) | 0 (0.0) | 1 (0.0) | |
| Daily total physical activity (MET-h/day), n (%) | | | | | |
| <16.3 | 203 (38.9) | 605 (36.4) | 593 (30.2) | 573 (24.9) | <0.001[d] |
| 16.3–28.2 | 175 (33.5) | 591 (35.5) | 700 (35.7) | 814 (35.4) | |
| ≥28.3 | 144 (27.6) | 467 (28.1) | 668 (34.1) | 910 (39.6) | |

(*Continued*)

**Table 1.** (Continued)

| | | | | | |
|---|---|---|---|---|---|
| Missing | 0 (0.0) | 0 (0.0) | 0 (0.0) | 2 (0.1) | |
| Leisure-time physical activity (MET-h/day), n (%) | | | | | |
| 0.0 | 145 (27.8) | 357 (21.5) | 385 (19.6) | 575 (25.0) | <0.001[d] |
| 0.1–2.1 | 202 (38.7) | 694 (41.7) | 857 (43.7) | 951 (41.4) | |
| ≥2.2 | 175 (33.5) | 612 (36.8) | 719 (36.7) | 771 (33.5) | |
| Missing | 0 (0.0) | 0 (0.0) | 0 (0.0) | 2 (0.1) | |
| The modified Japanese Diet Index (points)[e], n (%) | | | | | |
| <3.0 | 187 (35.8) | 490 (29.5) | 556 (28.4) | 743 (32.3) | 0.012[d] |
| 3.0–5.9 | 129 (24.7) | 419 (25.2) | 505 (25.8) | 564 (24.5) | |
| ≥6.0 | 206 (39.5) | 754 (45.3) | 900 (45.9) | 992 (43.2) | |
| Parity, n (%) | | | | | |
| 1 | 102 (19.5) | 218 (13.1) | 129 (6.6) | 15 (0.7) | <0.001[d] |
| 2 | 282 (54.0) | 1,047 (63.0) | 1,194 (60.9) | 528 (23.0) | |
| 3 | 105 (20.1) | 343 (20.6) | 548 (27.9) | 1,262 (54.9) | |
| ≥4 | 21 (4.0) | 42 (2.5) | 78 (4.0) | 483 (21.0) | |
| Missing | 12 (2.3) | 13 (0.8) | 12 (0.6) | 11 (0.5) | |
| Age at first delivery (years), n (%) | | | | | |
| <24 | 154 (29.5) | 467 (28.1) | 626 (31.9) | 959 (41.7) | <0.001[d] |
| 24–25 | 164 (31.4) | 581 (34.9) | 685 (34.9) | 739 (32.1) | |
| ≥26 | 190 (36.4) | 594 (35.7) | 624 (31.8) | 580 (25.2) | |
| Missing | 14 (2.7) | 21 (1.3) | 26 (1.3) | 21 (0.9) | |
| History of hormone replacement therapy, n (%) | | | | | |
| Never | 463 (88.7) | 1,468 (88.3) | 1,766 (90.1) | 2,106 (91.6) | 0.004[d] |
| Former or current | 51 (9.8) | 184 (11.1) | 180 (9.2) | 177 (7.7) | |
| Missing | 8 (1.5) | 11 (0.7) | 15 (0.8) | 16 (0.7) | |
| Body mass index (kg/m$^2$), mean ± SD | 22.8 ± 3.3 | 22.3 ± 3.0 | 22.7 ± 3.2 | 23.4 ± 3.3 | <0.001[c] |
| Systolic blood pressure (mm Hg), mean ± SD | 127.1 ± 17.6 | 126.8 ± 17.6 | 127.3 ± 17.7 | 128.0 ± 17.8 | 0.181[c] |
| Diastolic blood pressure (mm Hg), mean ± SD | 75.8 ± 10.2 | 76.1 ± 10.3 | 76.3 ± 10.3 | 76.3 ± 10.3 | 0.181[c] |
| Use of antihypertensive drugs, n (%) | | | | | |
| No | 401 (76.8) | 1,309 (78.7) | 1,502 (76.6) | 1,713 (74.5) | 0.022[d] |
| Yes | 121 (23.2) | 354 (21.3) | 459 (23.4) | 586 (25.5) | |
| Serum LDL-C (mg/dL), mean ± SD | 133.1 ± 33.2 | 130.9 ± 30.2 | 133.0 ± 30.6 | 132.8 ± 30.7 | 0.212[c] |
| Serum triglycerides (mg/dL), mean ± SD | 99.7 ± 48.8 | 97.3 ± 46.5 | 97.4 ± 46.9 | 98.3 ± 47.7 | 0.822[c] |
| Serum HDL-C (mg/dL), mean ± SD | 68.1 ± 16.7 | 70.0 ± 16.5 | 67.9 ± 16.2 | 66.5 ± 15.6 | <0.001[c] |
| Use of cholesterol-lowering drugs, n (%) | | | | | |
| No | 412 (78.9) | 1,312 (78.9) | 1,573 (80.2) | 1,878 (81.7) | 0.138[d] |
| Yes | 110 (21.1) | 351 (21.1) | 388 (19.8) | 421 (18.3) | |
| Blood glucose (mg/dL), mean ± SD | 95.8 ± 15.6 | 96.2 ± 16.0 | 96.3 ± 15.3 | 97.4 ± 17.1 | 0.049[c] |
| Use of antidiabetic drugs, n (%) | | | | | |
| No | 500 (95.8) | 1,611 (96.9) | 1,899 (96.8) | 2,184 (95.0) | 0.004[d] |
| Yes | 22 (4.2) | 52 (3.1) | 62 (3.2) | 115 (5.0) | |

Abbreviations: SD, standard deviation; MET-h/day, metabolic equivalent hours per day; LDL-C, low-density lipoprotein cholesterol; HDL-C, high-density lipoprotein cholesterol.

[a] Total breastfeeding duration was approximated as a product of the longest breastfeeding duration and the number of breastfed children and categorized into 0 and tertiles in participants with more than 0 months.

[b] Characteristics of participants were compared between groups of the total breastfeeding duration using the

[c] Kruskal-Wallis test for continuous variables and the

[d] chi-square test of independence for categorical variables.

[e] The modified Japanese Diet Index ranged from −2 to 9; higher score means higher quality of diet.

**Table 2. Associations of the longest breastfeeding duration, number of breastfed children, and total breastfeeding duration with metabolic syndrome in all participants and by age group.**

| Age group | Breastfeeding | | Participants (n) | Metabolic syndrome[a] n (%) | Age-adjusted OR (95% CI) | P trend[b] | Multivariable-adjusted[c] OR (95% CI) | P trend[b] |
|---|---|---|---|---|---|---|---|---|
| All participants | The longest breastfeeding duration (months)[d] | 0.0 | 686 | 95 (13.9) | 1.00 (reference) | 0.173 | 1.00 (reference) | 0.492 |
| | | 0.1–8.0 | 3,432 | 343 (10.0) | 0.80 (0.62 to 1.02) | | 0.86 (0.67 to 1.10) | |
| | | 8.1–12.0 | 3,906 | 450 (11.5) | 0.95 (0.75 to 1.21) | | 0.93 (0.73 to 1.19) | |
| | | ≥12.1 | 3,094 | 348 (11.3) | 0.95 (0.75 to 1.22) | | 0.96 (0.74 to 1.23) | |
| | Number of breastfed children[e] | 0 | 673 | 95 (14.1) | 1.00 (reference) | 0.006 | 1.00 (reference) | 0.699 |
| | | 1 | 1,327 | 136 (10.3) | 0.87 (0.66 to 1.16) | | 0.93 (0.68 to 1.27) | |
| | | 2 | 4,970 | 482 (9.7) | 0.78 (0.61 to 0.99) | | 0.83 (0.65 to 1.08) | |
| | | 3 | 3,307 | 378 (11.4) | 0.90 (0.71 to 1.16) | | 0.92 (0.67 to 1.27) | |
| | | ≥4 | 841 | 145 (17.2) | 1.45 (1.09 to 1.93) | | 1.21 (0.73 to 1.99) | |
| | Total breastfeeding duration (months)[f] | 0.0 | 686 | 95 (13.9) | 1.00 (reference) | 0.007 | 1.00 (reference) | 0.315 |
| | | 0.1–12.0 | 3,104 | 288 (9.3) | 0.74 (0.58 to 0.95) | | 0.81 (0.63 to 1.05) | |
| | | 12.1–30.0 | 3,263 | 357 (10.9) | 0.92 (0.72 to 1.17) | | 0.99 (0.77 to 1.27) | |
| | | ≥30.1 | 4,065 | 496 (12.2) | 1.01 (0.80 to 1.28) | | 0.96 (0.74 to 1.24) | |
| <55 years old | The longest breastfeeding duration (months)[d] | 0.0 | 164 | 22 (13.4) | 1.00 (reference) | 0.070 | 1.00 (reference) | 0.123 |
| | | 0.1–8.0 | 1,375 | 92 (6.7) | 0.45 (0.27 to 0.75) | | 0.49 (0.29 to 0.83) | |
| | | 8.1–12.0 | 1,691 | 131 (7.8) | 0.55 (0.34 to 0.90) | | 0.58 (0.35 to 0.97) | |
| | | ≥12.1 | 1,443 | 79 (5.5) | 0.41 (0.25 to 0.69) | | 0.45 (0.27 to 0.78) | |
| | Number of breastfed children[e] | 0 | 160 | 22 (13.8) | 1.00 (reference) | 0.466 | 1.00 (reference) | 0.006 |
| | | 1 | 644 | 36 (5.6) | 0.45 (0.25 to 0.79) | | 0.60 (0.31 to 1.17) | |
| | | 2 | 2,179 | 122 (5.6) | 0.39 (0.24 to 0.64) | | 0.50 (0.29 to 0.87) | |
| | | 3 | 1,339 | 104 (7.8) | 0.51 (0.31 to 0.84) | | 0.44 (0.24 to 0.84) | |
| | | ≥4 | 351 | 40 (11.4) | 0.74 (0.42 to 1.31) | | 0.35 (0.14 to 0.89) | |
| | Total breastfeeding duration (months)[f] | 0.0 | 164 | 22 (13.4) | 1.00 (reference) | 0.775 | 1.00 (reference) | 0.175 |
| | | 0.1–15.6 | 1,395 | 84 (6.0) | 0.43 (0.26 to 0.71) | | 0.49 (0.29 to 0.82) | |
| | | 15.7–32.0 | 1,534 | 104 (6.8) | 0.49 (0.30 to 0.81) | | 0.58 (0.35 to 0.98) | |
| | | ≥32.1 | 1,580 | 114 (7.2) | 0.51 (0.31 to 0.84) | | 0.48 (0.28 to 0.82) | |
| ≥55 years old | The longest breastfeeding duration (months)[d] | 0.0 | 522 | 73 (14.0) | 1.00 (reference) | 0.005 | 1.00 (reference) | 0.046 |
| | | 0.1–7.0 | 2,055 | 251 (12.2) | 0.88 (0.67 to 1.17) | | 0.97 (0.73 to 1.30) | |
| | | 7.1–12.0 | 2,217 | 319 (14.4) | 1.05 (0.80 to 1.38) | | 1.04 (0.78 to 1.37) | |
| | | ≥12.1 | 1,651 | 269 (16.3) | 1.20 (0.91 to 1.59) | | 1.20 (0.90 to 1.60) | |
| | Number of breastfed children[e] | 0 | 513 | 73 (14.2) | 1.00 (reference) | 0.016 | 1.00 (reference) | 0.358 |
| | | 1 | 683 | 100 (14.6) | 1.04 (0.75 to 1.45) | | 1.05 (0.74 to 1.48) | |
| | | 2 | 2,791 | 360 (12.9) | 0.91 (0.69 to 1.20) | | 0.94 (0.71 to 1.26) | |
| | | 3 | 1,968 | 274 (13.9) | 1.00 (0.76 to 1.33) | | 1.13 (0.78 to 1.64) | |
| | | ≥4 | 490 | 105 (21.4) | 1.67 (1.20 to 2.32) | | 1.82 (0.99 to 3.32) | |
| | Total breastfeeding duration (months)[f] | 0.0 | 522 | 73 (14.0) | 1.00 (reference) | 0.001 | 1.00 (reference) | 0.024 |
| | | 0.1–12.0 | 1,663 | 184 (11.1) | 0.79 (0.59 to 1.05) | | 0.88 (0.65 to 1.18) | |
| | | 12.1–30.0 | 1,961 | 286 (14.6) | 1.07 (0.81 to 1.42) | | 1.14 (0.86 to 1.52) | |
| | | ≥30.1 | 2,299 | 369 (16.1) | 1.18 (0.90 to 1.55) | | 1.17 (0.88 to 1.57) | |

Abbreviations: OR, odds ratio; CI, confidence interval; BMI, body mass index; HDL-C, high-density lipoprotein cholesterol; MET-h/day, metabolic equivalent hours per day.

[a] Metabolic syndrome was defined as satisfying at least three of the following five criteria: (a) obesity: BMI ≥25 kg/m$^2$; (b) elevated blood pressure: systolic blood pressure ≥130 mm Hg, diastolic blood pressure ≥85 mm Hg, and/or self-reported use of antihypertensive drugs; (c) elevated triglycerides: serum triglycerides ≥150 mg/dL; (d) reduced HDL-C: serum HDL-C <50 mg/dL; (e) elevated blood glucose: fasting blood glucose ≥100 mg/dL and/or self-reported use of antidiabetic drugs.

[b] P < 0.017 was considered statistically significant due to the Bonferroni correction.

[c] Adjusted for age (continuous), residential area (Tokai, Kinki, Shikoku or Kyushu), educational attainment (elementary or junior high school, high school or junior college, university or graduate school, missing), smoking status (never, former, current, missing), alcohol consumption (<55 years old: 0.0, 0.1–6.4, ≥6.5 g/day, missing; ≥55 years old: 0.0, 0.1–5.1, ≥5.2 g/day, missing), daily total physical activity (<55 years old: <14.0, 14.0–26.4, ≥26.5 MET-h/day, missing; ≥55 years old: <16.3, 16.3–28.2, ≥28.3 MET-h/day, missing), leisure-time physical activity (<55 years old: 0.0, 0.1–1.5, ≥1.6 MET-h/day, missing; ≥55 years old: 0.0, 0.1–2.1, ≥2.2 MET-h/day, missing), the modified Japanese Diet Index (<55 years old: <2.0, 2.0–4.9, ≥5.0 points, missing; ≥55 years old: <3.0, 3.0–5.9, ≥6.0 points, missing), parity (1, 2, 3, ≥4, missing), age at first delivery (<55 years old: <25, 25–27, ≥28 years old, missing; ≥55 years old: <24, 24–25, ≥26 years old, missing), and history of hormone replacement therapy (never, former, or current).

[d] The longest breastfeeding duration was categorized into 0 and tertiles in participants above 0 months.

[e] Number of breastfed children was defined as the number of children whom the participant had ever breastfed.

[f] Total breastfeeding duration was approximated and categorized in the same way as in Table 1.

the corresponding values were <0.001, 0.030, and 0.024, respectively, in the multivariable-adjusted model.

S1 Table shows associations between each measurement of breastfeeding history and metabolic syndrome when each measurement was incorporated as a continuous variable rather than as an ordinal variable into the models. The results remained almost the same as those of the trend tests in Table 2. The number of breastfed children was inversely associated with metabolic syndrome in the younger age group (< 55 years old), whereas the longest and total breastfeeding duration was positively correlated with the syndrome in the older age group (≥ 55 years old). In all participants, the longest and total breastfeeding duration was also correlated with the syndrome in this sensitivity analysis, which was different from the results of the main analysis with the tertiles. S2 Table shows associations between each measurement of breastfeeding history and metabolic syndrome when models incorporated only covariates that had preceded breastfeeding history in time. Although the point estimates and the confidence limits were lower, the results were materially unchanged from those in Table 2.

Table 3 shows associations between cardiovascular risk factors and the total breastfeeding duration in all participants and by age group. Inverse dose-response relationships were found for elevated LDL-C and dyslipidemia in all participants and those <55 years old and for elevated blood glucose in those <55 years old.

S3 Table shows the associations between BMI estimated using the self-reported weight at about 20 years of age and each measurement of breastfeeding history. BMI at about 20 years of age was positively associated with all indices of breastfeeding history, especially in all participants, and no inverse relationships were found.

## Discussion

In this cross-sectional study, associations between breastfeeding history and metabolic syndrome and cardiovascular risk factors were examined in parous women with stratification by age. In participants <55 years old, a history of breastfeeding their children was consistently associated with lower odds of metabolic syndrome compared with no such history. There were also inverse dose–response relationships between total breastfeeding duration and elevated LDL-C, dyslipidemia, and elevated blood glucose in all participants and those aged <55 years old.

Six previous studies found significant inverse associations between the duration of breastfeeding and metabolic syndrome [13–18], but four studies found no association [21–24]. This inconsistency may have been due to several factors. First, the participants' age could have affected the findings of the previous studies. Actually, in five of the former six studies that found inverse associations [13–17] but in only two of the latter four studies that found no association [22, 24], the mean or median age of the participants when measuring the outcome was younger than 50 years old. Furthermore, several studies and one systematic review reported that the protective effects of breastfeeding against hypertension, type 2 diabetes mellitus, cardiovascular risk factors, and cardiovascular disease could be limited to younger participants [25–28, 37]. Among them, the effect modification by age on the risks of hypertension, type 2 diabetes mellitus, and cardiovascular disease has been confirmed by a nested case-control study with prospectively assessed breastfeeding history [25] or by cohort studies [26, 28]. Taken together, these findings suggest that a longer interval after childbirth could attenuate the protective effects of breastfeeding on the risk of metabolic syndrome. Second, the median number of participants was 3,608 in the aforementioned six studies that found inverse associations between the duration of breastfeeding and metabolic syndrome [13–18] and was 952 in the aforementioned four studies that found no association [21–24], which should have led to

**Table 3. Associations between total breastfeeding duration and cardiovascular risk factors in all participants and by age group.**

| Risk factors | Age group | Total breastfeeding duration (weeks)[a] | Participants (n) | Outcomes[b] n (%) | Age-adjusted OR (95% CI) | P trend | Multivariable-adjusted[c] OR (95% CI) | P trend |
|---|---|---|---|---|---|---|---|---|
| Obesity | All participants | 0.0 | 686 | 155 (22.6) | 1.00 (reference) | <0.001 | 1.00 (reference) | 0.025 |
| | | 0.1–12.0 | 3,104 | 536 (17.3) | 0.76 (0.62 to 0.93) | | 0.81 (0.66 to 0.99) | |
| | | 12.1–30.0 | 3,263 | 623 (19.1) | 0.86 (0.71 to 1.05) | | 0.87 (0.71 to 1.07) | |
| | | ≥30.1 | 4,065 | 994 (24.5) | 1.17 (0.97 to 1.42) | | 1.02 (0.83 to 1.26) | |
| | <55 years old | 0.0 | 164 | 39 (23.8) | 1.00 (reference) | 0.065 | 1.00 (reference) | 0.410 |
| | | 0.1–15.6 | 1,395 | 240 (17.2) | 0.68 (0.46 to 1.00) | | 0.71 (0.48 to 1.06) | |
| | | 15.7–32.0 | 1,534 | 277 (18.1) | 0.73 (0.49 to 1.07) | | 0.72 (0.48 to 1.07) | |
| | | ≥32.1 | 1,580 | 335 (21.2) | 0.87 (0.60 to 1.28) | | 0.71 (0.47 to 1.06) | |
| | ≥55 years old | 0.0 | 522 | 116 (22.2) | 1.00 (reference) | <0.001 | 1.00 (reference) | 0.002 |
| | | 0.1–12.0 | 1,663 | 281 (16.9) | 0.72 (0.57 to 0.92) | | 0.80 (0.62 to 1.02) | |
| | | 12.1–30.0 | 1,961 | 404 (20.6) | 0.92 (0.73 to 1.16) | | 0.94 (0.74 to 1.20) | |
| | | ≥30.1 | 2,299 | 616 (26.8) | 1.29 (1.03 to 1.61) | | 1.18 (0.92 to 1.51) | |
| Elevated blood pressure | All participants | 0.0 | 686 | 331 (48.3) | 1.00 (reference) | 0.942 | 1.00 (reference) | 0.504 |
| | | 0.1–12.0 | 3,104 | 1,301 (41.9) | 0.99 (0.83 to 1.17) | | 1.03 (0.86 to 1.23) | |
| | | 12.1–30.0 | 3,263 | 1,329 (40.7) | 0.98 (0.82 to 1.16) | | 1.00 (0.84 to 1.20) | |
| | | ≥30.1 | 4,065 | 1,712 (42.1) | 0.99 (0.84 to 1.18) | | 0.98 (0.82 to 1.17) | |
| | <55 years old | 0.0 | 164 | 55 (33.5) | 1.00 (reference) | 0.166 | 1.00 (reference) | 0.053 |
| | | 0.1–15.6 | 1,395 | 364 (26.1) | 0.72 (0.51 to 1.03) | | 0.74 (0.51 to 1.06) | |
| | | 15.7–32.0 | 1,534 | 393 (25.6) | 0.72 (0.50 to 1.03) | | 0.74 (0.51 to 1.06) | |
| | | ≥32.1 | 1,580 | 403 (25.5) | 0.69 (0.48 to 0.98) | | 0.65 (0.45 to 0.94) | |
| | ≥55 years old | 0.0 | 522 | 276 (52.9) | 1.00 (reference) | 0.221 | 1.00 (reference) | 0.311 |
| | | 0.1–12.0 | 1,663 | 878 (52.8) | 1.04 (0.86 to 1.27) | | 1.10 (0.90 to 1.35) | |
| | | 12.1–30.0 | 1,961 | 1,036 (52.8) | 1.04 (0.85 to 1.26) | | 1.06 (0.87 to 1.30) | |
| | | ≥30.1 | 2,299 | 1,268 (55.2) | 1.11 (0.92 to 1.35) | | 1.15 (0.93 to 1.41) | |
| Elevated LDL-C[d] | All participants | 0.0 | 686 | 341 (49.7) | 1.00 (reference) | 0.002 | 1.00 (reference) | 0.002 |
| | | 0.1–12.0 | 3,104 | 1,300 (41.9) | 0.92 (0.77 to 1.09) | | 0.93 (0.78 to 1.11) | |
| | | 12.1–30.0 | 3,263 | 1,308 (40.1) | 0.88 (0.74 to 1.05) | | 0.87 (0.73 to 1.04) | |
| | | ≥30.1 | 4,065 | 1,592 (39.2) | 0.81 (0.68 to 0.96) | | 0.79 (0.66 to 0.95) | |
| | <55 years old | 0.0 | 164 | 52 (31.7) | 1.00 (reference) | 0.001 | 1.00 (reference) | <0.001 |
| | | 0.1–15.6 | 1,395 | 368 (26.4) | 0.80 (0.56 to 1.16) | | 0.83 (0.57 to 1.21) | |
| | | 15.7–32.0 | 1,534 | 354 (23.1) | 0.68 (0.47 to 0.98) | | 0.67 (0.46 to 0.98) | |
| | | ≥32.1 | 1,580 | 353 (22.3) | 0.62 (0.43 to 0.90) | | 0.60 (0.41 to 0.89) | |
| | ≥55 years old | 0.0 | 522 | 289 (55.4) | 1.00 (reference) | 0.301 | 1.00 (reference) | 0.906 |
| | | 0.1–12.0 | 1,663 | 872 (52.4) | 0.90 (0.74 to 1.10) | | 0.89 (0.73 to 1.09) | |
| | | 12.1–30.0 | 1,961 | 1,056 (53.9) | 0.95 (0.78 to 1.16) | | 0.96 (0.78 to 1.17) | |
| | | ≥30.1 | 2,299 | 1,197 (52.1) | 0.88 (0.73 to 1.06) | | 0.93 (0.75 to 1.14) | |
| Elevated triglycerides | All participants | 0.0 | 686 | 86 (12.5) | 1.00 (reference) | 0.363 | 1.00 (reference) | 0.324 |
| | | 0.1–12.0 | 3,104 | 318 (10.2) | 0.89 (0.69 to 1.15) | | 0.95 (0.73 to 1.23) | |
| | | 12.1–30.0 | 3,263 | 299 (9.2) | 0.80 (0.62 to 1.03) | | 0.88 (0.67 to 1.14) | |
| | | ≥30.1 | 4,065 | 408 (10.0) | 0.87 (0.68 to 1.11) | | 0.89 (0.68 to 1.16) | |
| | <55 years old | 0.0 | 164 | 19 (11.6) | 1.00 (reference) | 0.232 | 1.00 (reference) | 0.244 |
| | | 0.1–15.6 | 1,395 | 100 (7.2) | 0.61 (0.36 to 1.02) | | 0.67 (0.39 to 1.15) | |
| | | 15.7–32.0 | 1,534 | 97 (6.3) | 0.54 (0.32 to 0.90) | | 0.64 (0.37 to 1.12) | |
| | | ≥32.1 | 1,580 | 110 (7.0) | 0.58 (0.35 to 0.98) | | 0.62 (0.35 to 1.09) | |
| | ≥55 years old | 0.0 | 522 | 60 (12.8) | 1.00 (reference) | 0.822 | 1.00 (reference) | 0.990 |
| | | 0.1–12.0 | 1,663 | 218 (12.7) | 0.96 (0.71 to 1.29) | | 1.04 (0.77 to 1.41) | |
| | | 12.1–30.0 | 1,961 | 200 (12.0) | 0.90 (0.67 to 1.21) | | 0.99 (0.73 to 1.33) | |
| | | ≥30.1 | 2,299 | 225 (12.5) | 0.96 (0.72 to 1.27) | | 1.03 (0.76 to 1.40) | |

*(Continued)*

**Table 3.** (Continued)

| Risk factors | Age group | Total breastfeeding duration (weeks)[a] | Participants (n) | Outcomes[b] n (%) | Age-adjusted OR (95% CI) | P trend | Multivariable-adjusted[c] OR (95% CI) | P trend |
|---|---|---|---|---|---|---|---|---|
| Reduced HDL-C | All participants | 0.0 | 686 | 79 (11.5) | 1.00 (reference) | 0.029 | 1.00 (reference) | 0.383 |
| | | 0.1–12.0 | 3,104 | 284 (9.2) | 0.85 (0.65 to 1.10) | | 0.94 (0.72 to 1.24) | |
| | | 12.1–30.0 | 3,263 | 316 (9.7) | 0.91 (0.70 to 1.19) | | 1.03 (0.78 to 1.35) | |
| | | ≥30.1 | 4,065 | 455 (11.2) | 1.06 (0.82 to 1.37) | | 1.04 (0.79 to 1.37) | |
| | <55 years old | 0.0 | 164 | 18 (11.0) | 1.00 (reference) | 0.657 | 1.00 (reference) | 0.346 |
| | | 0.1–15.6 | 1,395 | 120 (8.6) | 0.77 (0.45 to 1.30) | | 0.92 (0.53 to 1.59) | |
| | | 15.7–32.0 | 1,534 | 109 (7.1) | 0.63 (0.37 to 1.06) | | 0.79 (0.45 to 1.37) | |
| | | ≥32.1 | 1,580 | 137 (8.7) | 0.77 (0.46 to 1.30) | | 0.82 (0.46 to 1.44) | |
| | ≥55 years old | 0.0 | 522 | 61 (11.7) | 1.00 (reference) | 0.001 | 1.00 (reference) | 0.074 |
| | | 0.1–12.0 | 1,663 | 144 (8.7) | 0.75 (0.54 to 1.03) | | 0.86 (0.62 to 1.19) | |
| | | 12.1–30.0 | 1,961 | 239 (12.2) | 1.09 (0.80 to 1.47) | | 1.17 (0.86 to 1.60) | |
| | | ≥30.1 | 2,299 | 306 (13.3) | 1.17 (0.87 to 1.57) | | 1.12 (0.81 to 1.54) | |
| Dyslipidemia | All participants | 0.0 | 686 | 396 (57.7) | 1.00 (reference) | 0.023 | 1.00 (reference) | 0.007 |
| | | 0.1–12.0 | 3,104 | 1,519 (48.9) | 0.88 (0.74 to 1.04) | | 0.91 (0.76 to 1.08) | |
| | | 12.1–30.0 | 3,263 | 1,550 (47.5) | 0.86 (0.72 to 1.02) | | 0.87 (0.73 to 1.04) | |
| | | ≥30.1 | 4,065 | 1,923 (47.3) | 0.82 (0.69 to 0.97) | | 0.80 (0.67 to 0.96) | |
| | <55 years old | 0.0 | 164 | 65 (39.6) | 1.00 (reference) | 0.003 | 1.00 (reference) | 0.002 |
| | | 0.1–15.6 | 1,395 | 478 (34.3) | 0.83 (0.59 to 1.16) | | 0.88 (0.62 to 1.24) | |
| | | 15.7–32.0 | 1,534 | 454 (29.6) | 0.67 (0.48 to 0.94) | | 0.70 (0.49 to 0.99) | |
| | | ≥32.1 | 1,580 | 484 (30.6) | 0.68 (0.48 to 0.96) | | 0.67 (0.47 to 0.97) | |
| | ≥55 years old | 0.0 | 522 | 331 (63.4) | 1.00 (reference) | 0.759 | 1.00 (reference) | 0.951 |
| | | 0.1–12.0 | 1,663 | 977 (58.8) | 0.84 (0.69 to 1.03) | | 0.86 (0.70 to 1.06) | |
| | | 12.1–30.0 | 1,961 | 1,214 (61.9) | 0.96 (0.78 to 1.17) | | 0.98 (0.79 to 1.20) | |
| | | ≥30.1 | 2,299 | 1,385 (60.2) | 0.88 (0.72 to 1.07) | | 0.90 (0.73 to 1.11) | |
| Elevated blood glucose | All participants | 0.0 | 686 | 186 (27.1) | 1.00 (reference) | 0.920 | 1.00 (reference) | 0.364 |
| | | 0.1–12.0 | 3,104 | 648 (20.9) | 0.83 (0.69 to 1.01) | | 0.84 (0.69 to 1.02) | |
| | | 12.1–30.0 | 3,263 | 662 (20.3) | 0.82 (0.68 to 1.00) | | 0.83 (0.68 to 1.01) | |
| | | ≥30.1 | 4,065 | 898 (22.1) | 0.89 (0.74 to 1.08) | | 0.85 (0.70 to 1.04) | |
| | <55 years old | 0.0 | 164 | 36 (22.0) | 1.00 (reference) | 0.028 | 1.00 (reference) | 0.025 |
| | | 0.1–15.6 | 1,395 | 186 (13.3) | 0.56 (0.37 to 0.85) | | 0.59 (0.39 to 0.90) | |
| | | 15.7–32.0 | 1,534 | 206 (13.4) | 0.58 (0.38 to 0.87) | | 0.63 (0.41 to 0.97) | |
| | | ≥32.1 | 1,580 | 195 (12.3) | 0.51 (0.34 to 0.76) | | 0.50 (0.32 to 0.78) | |
| | ≥55 years old | 0.0 | 522 | 150 (28.7) | 1.00 (reference) | 0.079 | 1.00 (reference) | 0.274 |
| | | 0.1–12.0 | 1,663 | 427 (25.7) | 0.87 (0.70 to 1.08) | | 0.86 (0.68 to 1.07) | |
| | | 12.1–30.0 | 1,961 | 516 (26.3) | 0.89 (0.72 to 1.11) | | 0.89 (0.71 to 1.11) | |
| | | ≥30.1 | 2,299 | 678 (29.5) | 1.04 (0.84 to 1.28) | | 1.01 (0.81 to 1.27) | |

Abbreviations: OR, odds ratio; CI, confidence interval; LDL-C, low-density lipoprotein cholesterol; HDL-C, high-density lipoprotein cholesterol.

[a] Total breastfeeding duration was approximated and categorized in the same way as in Table 1.

[b] Elevated LDL-C was defined as serum LDL-C ≥140 mg/dL and/or use of cholesterol-lowering drugs, and dyslipidemia was defined in those who satisfied at least one of the following three criteria: elevated LDL-C, elevated triglycerides, and reduced HDL-C. The definitions of the other outcomes were the same as those in Table 2.

[c] Models incorporated the same covariates as those of multivariable-adjusted models in Table 2.

[d] Serum concentration of LDL-C was calculated using the Friedewald formula (LDL-C = total cholesterol − HDL-C − triglycerides/5).

higher statistical power in the former. Third, the differences in the proportions of participants with breastfeeding history and metabolic syndrome may have led to the differences in the findings. Yet, the median proportions of participants with breastfeeding history and of those with metabolic syndrome were 74.6% and 19.2% in the former six studies, and 75.1% and 26.5%, respectively, in the latter four studies, and thus their differences did not seem to be large. Fourth, study design may have influenced the findings of the previous studies. However, only one cohort study was included in both the former six [14] and the latter four studies [24], and all the remaining studies were cross-sectional studies. Cohort studies including older participants are needed to confirm the effects of breastfeeding history on metabolic syndrome.

Little epidemiological and mechanistic knowledge is available about the effects of breastfeeding history on the risk of metabolic syndrome in older women. However, high parity may be a risk factor for metabolic syndrome and diabetes mellitus [15, 38–40]. Although parity was adjusted in our multivariable models, our estimates of exposure, which were the number of breastfed children or the total breastfeeding duration, possibly included residual confounding from parity, which can bias the association in an upward direction. Additionally, selection bias may distort the estimates in a positive direction. No breastfeeding history or a shorter duration of breastfeeding can increase the risks of breast and endometrial cancers [8, 9] and cardiovascular disease [19], and metabolic syndrome can also increase the risks of these diseases [2, 41, 42]. If our group of older women without breastfeeding history excludes more potential participants who have developed these diseases than other groups because of their death or a decline in their health status, selection bias could have affected the present estimates [43].

Regarding cardiovascular risk factors, which were the secondary outcomes, a meta-analysis of cohort studies has demonstrated an inverse dose–response relationship between the total breastfeeding duration and the risk of type 2 diabetes mellitus [10], and a meta-analysis of observational studies showed that breastfeeding history was associated with lower risks or odds of hypertension compared with no breastfeeding history [11]. Furthermore, a few cross-sectional studies found that the total breastfeeding duration was inversely associated with serum lipid levels, especially in younger participants [27, 44]. The present results examining cardiovascular risk factors generally agreed with the findings of these previous studies [10, 27, 44] and supported the effect modification by age on the associations between breastfeeding history and cardiovascular risk factors suggested by several studies [25–28, 44]. Yet, the multiple comparisons of cardiovascular risk factors could have led to an inflated type I error, and thus, caution should be exercised when interpreting the present results.

Several biological mechanisms could explain the inverse associations between breastfeeding and the risks of metabolic syndrome and cardiovascular risk factors in younger participants. First, circulating levels of the pituitary hormone, prolactin, increase during pregnancy and lactation [45], and a recent study confirmed that prolactin levels are inversely associated with the risk of type 2 diabetes mellitus, even with 22 years of follow-up [46]. Second, breastfeeding induces the release of oxytocin [47], which exerts effects that could lower blood pressure and reduce psychological stress [48]. Chronic oxytocin exposure may also lead to reduced body weight and fat and improve glucose homeostasis [49]. Third, exclusive breastfeeding requires mothers to consume an extra 400–600 kcal/day, and 150–170 kcal/day of them are mobilized from tissue stores [50]. Fourth, breastfeeding has been associated with increased levels of adipokines such as ghrelin and peptide YY and these cytokines could reduce risks of diabetes and obesity [51]. Fifth, apolipoprotein D is involved in lipid transport [52]. Its level decreases during pregnancy but it returns to baseline levels during breastfeeding [53], which may decrease triglyceride levels [54].

The strengths of the present study include a relatively large number of community-dwelling participants, extensive adjustments for potential confounding factors, inclusion of older

participants, and stratified analyses by age. On the other hand, eight limitations should be mentioned. First, breastfeeding histories were assessed using self-reports, which may have included measurement errors. However, recalled breastfeeding duration agrees well with prospectively assessed data collected 20 years after delivery [55], and agrees moderately well, even in participants aged 69–79 years [56]. Second, total breastfeeding duration was approximated by the product of the number of breastfed children and the longest breastfeeding duration, but this could have included some misclassification because breastfeeding duration may have varied among children from the same mother. The total breastfeeding duration of the present study probably overestimated the true duration due to using the longest breastfeeding duration in the calculation. It is difficult to ascertain the direction of the bias derived from this overestimation [57]. A national survey in Japan reported no large differences in the proportion of breastfed children between birth orders six months after delivery [58], but the proportions may have differed among the birth orders in later months. Third, our questionnaire did not have items concerning exclusive breastfeeding, which could have produced misclassification in the intensity of breastfeeding. Nevertheless, few previous studies have demonstrated that exclusive breastfeeding is more effective than partial breastfeeding in reducing the maternal risks of metabolic syndrome or cardiovascular risk factors [10–18, 21–24, 59]. Future studies should conduct an analysis that evaluates effects of exclusive and partial breastfeeding separately. Fourth, a systematic review showed that obese women are less likely to initiate and maintain breastfeeding for a long time [30], which could lead to reverse causality. However, the additional analyses found rather positive correlations between the body mass index at about 20 years of age and each measurement of breastfeeding history. Fifth, although evidence is insufficient [60], several observational studies showed that women with gestational diabetes are also less likely to initiate and maintain breastfeeding [60, 61], which is another potential cause of reverse causality. Sixth, multiple comparisons of cardiovascular risk factors should have produced false-positive results, and thus, these analyses should be regarded as exploratory rather than confirmatory. Seventh, the cross-sectional design of the present study prevented exclusion of recall bias as well as reverse causality. Participants without metabolic syndrome may have more accurately recalled their breastfeeding history than participants with metabolic syndrome due to their higher level of health consciousness, which could have biased the estimates in a negative direction. Eighth, some covariates adjusted in the present study followed breastfeeding history in time, and thus the adjustment for these potential mediators may have diluted the associations. Still, the results of the sensitivity analysis that incorporated only covariates that had preceded breastfeeding history in time were materially unchanged from those of the analysis that retained all the covariates (S2 Table). Additionally, there is no evidence that breastfeeding history affects the covariates incorporated into a multivariable-adjusted model long after delivery. Considering these points, the adjustment for the covariates was less likely to have profoundly biased the estimates in the present study. Furthermore, if the covariates of the present study are affected by unmeasured confounding variables and the effects of confounding are strong, not adjusting for these covariates may lead to residual confounding even if these covariates are also mediators [43, 62]. A simulation study recommends presenting both results adjusted and unadjusted for potential mediators that are also descendants of confounding variables when only variables that concede exposure were measured and sufficient knowledge is not available about relationships between variables examined in studies [62]. Following this recommendation, we have retained all the covariates in main models (Tables 2 and 3) and conducted the sensitivity analysis mentioned above (S2 Table). To obviate these eight limitations, cohort studies with prospective assessment of breastfeeding history for each child, metabolic syndrome, BMI before pregnancy, gestational complications and other potential confounding factors should be conducted.

In conclusion, breastfeeding history may be related to lower prevalence of metabolic syndrome in middle-aged parous women. Because many women deliver a baby, promotion of breastfeeding by parous women could have impacts at the population level on prevention of metabolic syndrome and its sequelae including cardiovascular disease. However, attention should be paid when interpreting the findings of the present analysis. In addition, cohort studies are warranted because of limitations due to the cross-sectional design of the current study.

## Supporting information

**S1 Table. Associations between each measurement of breastfeeding history (incorporated as continuous variables) and metabolic syndrome in all participants and by age group.** Abbreviations: OR, odds ratio; CI, confidence interval. [a, b, c] Each index of breastfeeding history was measured in the same way as those in Tables 1 and 2 and was incorporated in the models as a continuous variable. [d] This model incorporated the same covariates as those of the multivariable-adjusted model in Table 2.
(XLSX)

**S2 Table. Associations between each measurement of breastfeeding history and metabolic syndrome in all participants and by age group, when incorporating only covariates that preceded breastfeeding history in time.** Abbreviations: OR, odds ratio; CI, confidence interval. [a, b, c] Each index of breastfeeding history was measured and categorized in the same way as those in Tables 1 and 2. [d] Adjusted for age, residential area, educational attainment, age at first delivery, and parity. The categories of these covariates were the same as those in Table 2.
(XLSX)

**S3 Table. Associations between body mass index estimated using the self-reported body weight at about 20 years of age and each index of breastfeeding history.** Abbreviations: OR, odds ratio; CI, confidence interval. [a] All the indices of breastfeeding history were defined and categorized in the same way as in Tables 1 and 2. [b] Higher categories of breastfeeding history were as follows in all participants, those <55 years old, and those ≥55 years old: the longest breastfeeding duration: ≥8.1, ≥8.1, and ≥7.1 months, respectively; number of breastfed children: ≥2 children in all age groups; and total breastfeeding duration: ≥12.1, ≥15.7, and ≥12.1 months, respectively. [c] Models incorporated the same covariates as those of multivariable-adjusted models in Table 2 in the analyses of the longest and the total breastfeeding duration. In the analyses of the number of breastfed children, models incorporated the same covariates except for parity. [d] Odds ratios and 95% CIs were calculated using the unconditional binary logistic regression model.
(XLSX)

## Acknowledgments

The authors wish to thank the previous principal investigators of the J-MICC Study, Drs. Nobuyuki Hamajima and Hideo Tanaka, for their efforts in the establishment and follow-up of the cohort.

## Author Contributions

**Conceptualization:** Takashi Matsunaga, Kenji Wakai.

**Data curation:** Yuka Kadomatsu, Mineko Tsukamoto, Yoko Kubo, Rieko Okada, Mako Nagayoshi, Takashi Tamura, Asahi Hishida, Toshiro Takezaki, Ippei Shimoshikiryo, Sadao Suzuki, Hiroko Nakagawa, Naoyuki Takashima, Yoshino Saito, Kiyonori Kuriki, Kokichi

Arisawa, Sakurako Katsuura-Kamano, Nagato Kuriyama, Daisuke Matsui, Haruo Mikami, Yohko Nakamura, Isao Oze, Hidemi Ito, Masayuki Murata, Hiroaki Ikezaki, Yuichiro Nishida, Chisato Shimanoe, Kenji Takeuchi.

**Formal analysis:** Takashi Matsunaga.

**Writing – original draft:** Takashi Matsunaga, Kenji Wakai.

**Writing – review & editing:** Kenji Wakai.

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
