## [Decision Letter · Decision Letter 0]

22 Feb 2021

PONE-D-21-02465

Associations of breastfeeding history with metabolic syndrome and cardiovascular risk factors in community-dwelling parous women: The Japan Multi-Institutional Collaborative Cohort Study

PLOS ONE

Dear Dr. Matsunaga,

Thank you for submitting your manuscript to PLOS ONE. After careful consideration, we feel that it has merit but does not fully meet PLOS ONE’s publication criteria as it currently stands. Therefore, we invite you to submit a revised version of the manuscript that addresses the points raised during the review process.

We look forward to receiving your revised manuscript.

Kind regards,

Antonio Palazón-Bru, PhD

Academic Editor

PLOS ONE

Journal Requirements:

2. Please refer to the specific statistical analyses performed as well as any post-hoc corrections to correct for multiple comparisons. If these were not performed please justify the reasons. Please refer to our statistical reporting guidelines for assistance (https://journals.plos.org/plosone/s/submission-guidelines.#loc-statistical-reporting). Additionally, please ensure you have thoroughly discussed any potential limitations of this study within the Discussion.

Reviewers' comments:

Reviewer's Responses to Questions

**Comments to the Author**

1. Is the manuscript technically sound, and do the data support the conclusions?

Reviewer #1: Partly

Reviewer #2: Partly

Reviewer #3: Yes

2. Has the statistical analysis been performed appropriately and rigorously? 

Reviewer #1: Yes

Reviewer #2: Yes

Reviewer #3: No

3. Have the authors made all data underlying the findings in their manuscript fully available?

Reviewer #1: No

Reviewer #2: Yes

Reviewer #3: No

4. Is the manuscript presented in an intelligible fashion and written in standard English?

Reviewer #1: Yes

Reviewer #2: Yes

Reviewer #3: Yes

5. Review Comments to the Author

Reviewer #1: This study aims to investigate the associations between breastfeeding history and the prevalence of metabolic syndrome and cardiovascular risk factors in parous women. The paper is well written and investigates an interesting topic. However, the main concern is that the conclusion does not seem justified and supported by the results. Specifically, the reported associations does not seem to be significant.

Reviewer #2: OVERALL COMMENTS TO THE AUTHORS

This study examines the association between breastfeeding and the prevalence of metabolic syndrome and cardiovascular risk factors in community-dwelling parous women, and includes stratified analyses by age. This is an interesting research area. The study is thorough in its analysis, and is well written. However I mainly had concerns with the Discussion (please see my specific comments) and interpretation of findings given the cross-sectional nature of the study.

SPECIFIC COMMENTS TO THE AUTHORS

Introduction

Lines 95-98 Is there more recent data that can be referenced than reference 7?

Results

Line 235 Was presenting breastfeeding in terms of months considered rather than weeks? Weeks may be less intuitive to relate to in terms of breastfeeding duration, as breastfeeding recommendations are usually expressed in months.

Discussion

Generally, the discussion was very descriptive in various instances but could benefit from being more concise overall.

Line 318 I would suggest being more careful with interpretation and language used (e.g. use "may have" rather than "probably have"). There could be other factors besides age to be considered and discussed, and that can explain some of the differences observed such as sample size, sample characteristics, follow-up periods, etc.

Lines 319-324 The findings from the study need to be discussed in light of the evidence provided by other studies and a comment on the study design of these studies should be included. How much of the evidence was from longitudinal/cross-sectional studies?

Lines 324-326 Again, did this evidence come from mainly longitudinal or cross-sectional studies? Has there been any systematic review examining whether age affects associations of breastfeeding with cardiovascular risk factors or cardiovascular disease?

Study limitations

What about the limitation of the breastfeeding variable which was a product of the number of breastfed children and longest breastfeeding duration? Breastfeeding duration may have varied among children from the same mother, and the breastfeeding duration used in the study may not accurately reflect the true overall breastfeeding duration across children. Please add this limitation/discuss.

Other comments

Acknowledging the limitations of cross-sectional evidence, what future studies are needed for further inform the evidence base and extend these findings? Please add this to the discussion.

Conclusions

The conclusions should remind readers that this study provides evidence of a cross-sectional nature, which limits inferences and statements that are too conclusive. Please reword accordingly.

In addition, please add that this is a cross-sectional study in the Methods of the Abstract.

Reviewer #3: This is an article with a very interesting theme. However, I believe that some points should be considered for possible publication.

Minor comments:

Abstract:

The abstract needs review. The purpose of the abstract should be the same as that described at the end of the introduction. In my opinion, the abstract is more correct according to the study. In the methods, it is also important to highlight the breastfeeding categories already in this part of the manuscript. The abstract is long, but the results incomplete. The authors mention the sample starting at 10,432, which creates confusion. Still in the results I suggest that the way of presenting the data with measure of effect (OR) and their respective 95% CI should be as follows, for example: [OR = 0.57 (95% CI 0.36; 0.89)]. This comment applies to the abstract and all text. The measure of effect used in the study, obtained through logistic regression, was an odds ratio (ratio). Therefore, the interpretation of the results mentioning lower prevalence is totally wrong. In this case it would be "lower odds". (see line 78 of the abstract).

Introduction

Review carefully so that the abstract objective and introduction are compatible.

Material and Methods

It would be interesting for the authors to evaluate the time of exclusive breastfeeding separately from the time of total breastfeeding. There is evidence that shows the influence of EBF on cardiovascular outcomes, for example. I suggest the following reference: https://www.nature.com/articles/s41366-018-0317-5.

Line 214: Why is this test not a linear regression? Didn't outcome and exposure have a normal distribution?

Line 226: Even if it was the median age of the study, it does not characterize a strong justification regarding the stratification of the sample. I strongly suggest revising. Maybe it would be better to use more usual maternal age stratifications, separating younger and older mothers.

Line 235: This categorization of breastfeeding weeks is too confusing even to refer to the results of the study.

Results

Table 1. It was not mentioned in the methods that there would be a description of the sample according to the breastfeeding categories. Indicate at the bottom of the table which test is used for p-value.

Table 2. I confess that this variable (Total breastfeeding duration weeks- Total breastfeeding duration was approximated as a product of the longest breastfeeding duration and the number of breastfed children) is still confused. The authors have enough information, this variable hinders more than helps in the interpretation of results.

Discussion

Line 417: "Finally, the cross-sectional design of our study prevented exclusion of recall bias and reverse causality". Why? Need to justify this claim.

The authors must present a figure with the study sample flowchart, thus facilitating the understanding of the entire sample selection.

The number of references is extravagant! I suggest reviewing and reducing.

Major comments

Authors need to correctly review the analysis model. This part is very sensitive in this work, it leaves some doubts about the entry of variables in the model. My concern is that they are adjusting for mediator in the wrong way. In addition, you need to check the need for all the variables presented. Authors need to consider the idea of creating a variable of exclusive breastfeeding and total breastfeeding, there are differences. Still, to justify the etsratifications performed (maternal age, for example) the authors need to test and present the findings for interaction.

6. PLOS authors have the option to publish the peer review history of their article (what does this mean?). If published, this will include your full peer review and any attached files.

Reviewer #1: No

Reviewer #2: No

Reviewer #3: No

---

## [Author Response · Author response to Decision Letter 0]

5 Jun 2021

June 5, 2021

Antonio Palazón-Bru, Ph.D.

Academic Editor

PLOS ONE

Dear Dr. Palazón-Bru,

Thank you for inviting us to submit our revised manuscript entitled, “Associations of breastfeeding history with metabolic syndrome and cardiovascular risk factors in community-dwelling parous women: The Japan Multi-Institutional Collaborative Cohort Study” to PLOS ONE (Reference Number: PONE-D-21-02465). We also appreciate the time and effort you and the reviewers have dedicated to providing feedback. We have incorporated modifications that reflect the suggestions you have provided. We hope that our edits and the responses we provide below satisfactorily address all the issues and concerns you and the reviewers have noted.

To facilitate your review of our revisions, the following are point-by-point responses to the questions and comments delivered in your letter dated February 23, 2021. Amendments that we have made are shown in colored text throughout the Revised Article with Track Changes.

Academic Editor’s comments

Response: We have embedded the editable tables and inserted the captions of the tables and a figure in the text of the manuscript. We have also named our files according to the journal’s requirements.

(2) Please refer to the specific statistical analyses performed as well as any post-hoc corrections to correct for multiple comparisons. If these were not performwed please justify the reasons.…Additionally, please ensure you have thoroughly discussed any potential limitations of this study within the Discussion.

Response: In the revised manuscript, we have mentioned that our primary outcome was metabolic syndrome and secondary outcomes were cardiovascular risk factors (lines 65–67 and 168–169 in the unmarked manuscript). We conducted Bonferroni corrections for multiple comparisons in the trend tests examining the association between breastfeeding history and metabolic syndrome, because we used three indices of breastfeeding history (the longest breastfeeding duration, the number of breastfed children, and the total breastfeeding duration). Regarding the tests examining the association between categorized breastfeeding history and metabolic syndrome, we have not conducted any corrections for multiple comparisons because observational epidemiological studies rarely conduct these corrections on such analyses (lines 266–276). In the Discussion, we have added potential limitations of the present study including multiple comparisons as well as potential measurement errors and misclassification in breastfeeding history, reverse causality, and recall bias (lines 464–466 and 493–513).

(3) We note that you have included the phrase “data not shown” in your manuscript. Unfortunately, this does not meet our data sharing requirements. PLOS does not permit references to inaccessible data. We require that authors provide all relevant data within the paper, Supporting Information files, or in an acceptable, public repository.

Response: We have added a supplementary table (S1 Table) reporting the associations between body mass index at about 20 years old and breastfeeding history and have mentioned the methods of statistical analysis (lines 243–265) and results (lines 399–402) in the revised manuscript.

(4) We note that you have indicated that data from this study are available upon request. PLOS only allows data to be available upon request if there are legal or ethical restrictions on sharing data publicly.

Response: The data used in the present study cannot be made publicly available because the participants of the present study have not given informed consent for public data sharing and the ethics committee of the Nagoya University Graduate School of Medicine does not approve the sharing. To obtain access to the confidential data, interested researchers can contact the Administration Office of the ethics committee of the Nagoya University Graduate School of Medicine (iga-shinsa@adm.nagoya-u.ac.jp). After receiving a request, informed consent from the participants will be sought.

Reviewer #1’s comment

(1) the main concern is that the conclusion does not seem justified and supported by the results. Specifically, the reported associations do not seem to be significant.

Response: Our results consistently indicated that breastfeeding history may be inversely related to the prevalence of metabolic syndrome and cardiovascular risk factors, especially in participants <55 years old. Even after Bonferroni corrections, the number of breastfed children was significantly associated with lower prevalence of metabolic syndrome in participants <55 years old.

Reviewer #2’s comments

(1) Lines 95-98 Is there more recent data that can be referenced than reference 7?

Response: More recent analyses using the data from the Global Burden of Disease have not reported the combined effect of high blood pressure, blood glucose, serum cholesterol, and body mass index on death from cardiovascular disease, chronic kidney disease, and diabetes (Lancet. 2017;390(10100):1345-1422. and Lancet. 2018;392(10159):1923-1994.) Thus, we have retained the article cited in the previous manuscript (lines 92–95).

(2) Line 235 Was presenting breastfeeding in terms of months considered rather than weeks? Weeks may be less intuitive to relate to in terms of breastfeeding duration, as breastfeeding recommendations are usually expressed in months.

Response: We have changed our breastfeeding duration measured in weeks to that measured in months throughout the manuscript including the tables.

(3) Generally, the discussion was very descriptive in various instances but could benefit from being more concise overall.

Response: In the revised manuscript, we have reduced the descriptions about cardiovascular risk factors because they were the secondary outcomes in the present study, and our analysis provided generally consistent results with previous studies about these risk factors (lines 378–381 and lines 454–466).

(4) Line 318 I would suggest being more careful with interpretation and language used (e.g. use "may have" rather than "probably have"). There could be other factors besides age to be considered and discussed, and that can explain some of the differences observed such as sample size, sample characteristics, follow-up periods, etc.

(5) Lines 319-324 The findings from the study need to be discussed in light of the evidence provided by other studies and a comment on the study design of these studies should be included. How much of the evidence was from longitudinal/cross-sectional studies?

Response: According to the reviewers’ recommendation, we have corrected the language about the influence of age (lines 414–415). Furthermore, we have explained the differences in sample size, the proportions of participants with breastfeeding history and metabolic syndrome, and study design between our study and previous studies (lines 427–440).

(6) Lines 324-326 Again, did this evidence come from mainly longitudinal or cross-sectional studies? Has there been any systematic review examining whether age affects associations of breastfeeding with cardiovascular risk factors or cardiovascular disease?

Response: In the revised manuscript, we have mentioned which associations come from case-control or cohort studies and that no systematic review or meta-analysis has been conducted about the effect modification by age on the associations of breastfeeding history with cardiovascular risk factors or cardiovascular disease (lines 418–425).

(7) What about the limitation of the breastfeeding variable which was a product of the number of breastfed children and longest breastfeeding duration? Breastfeeding duration may have varied among children from the same mother, and the breastfeeding duration used in the study may not accurately reflect the true overall breastfeeding duration across children. Please add this limitation/discuss.

Response: We have added the potential limitation of how we calculated total breastfeeding duration in the Discussion of the revised manuscript (lines 498–501).

(8) Acknowledging the limitations of cross-sectional evidence, what future studies are needed for further inform the evidence base and extend these findings? Please add this to the discussion.

(9) The conclusions should remind readers that this study provides evidence of a cross-sectional nature, which limits inferences and statements that are too conclusive. Please reword accordingly.

Response: In the Discussion of the revised manuscript, we have added what studies should be conducted in the future to clarify the associations between breastfeeding history and metabolic syndrome (lines 511–513), and have modified the conclusion (lines 514–520).

(10) Please add that this is a cross-sectional study in the Methods of the Abstract.

Response: We have stated that our study is a cross-sectional design in the Methods of the Abstract in the revised manuscript (line 60).

Reviewer #3’s comments

(1) The abstract needs review. The purpose of the abstract should be the same as that described at the end of the introduction. In my opinion, the abstract is more correct according to the study. In the methods, it is also important to highlight the breastfeeding categories already in this part of the manuscript. The abstract is long, but the results incomplete. The authors mention the sample starting at 10,432, which creates confusion. Still in the results I suggest that the way of presenting the data with measure of effect (OR) and their respective 95% CI should be as follows, for example: [OR = 0.57 (95% CI 0.36; 0.89)]. This comment applies to the abstract and all text. The measure of effect used in the study, obtained through logistic regression, was an odds ratio (ratio). Therefore, the interpretation of the results mentioning lower prevalence is totally wrong. In this case it would be "lower odds". (see line 78 of the abstract).

(2) Review carefully so that the abstract objective and introduction are compatible.

Response: We have made the following revisions. First, we have modified the Objective of the Abstract to include examining the effect modification by age to align the aim with that in the Introduction (lines 56–58). Second, we have explained the categories of breastfeeding duration in the Methods section of the Abstract (lines 64–65). Third, we have again indicated that our total sample size was 11,118 women in the Results section of the Abstract (lines 71–72). Fourth, we have revised the descriptions about the results of logistic regression analysis throughout the revised manuscript to correct the interpretation of the results.

(3) It would be interesting for the authors to evaluate the time of exclusive breastfeeding separately from the time of total breastfeeding. There is evidence that shows the influence of EBF on cardiovascular outcomes, for example. I suggest the following reference: https://www.nature.com/articles/s41366-018-0317-5.

Response: Unfortunately, our questionnaire did not have items about exclusive breastfeeding, and thus, we could not separate breastfeeding duration into exclusive and partial durations. Moreover, no previous studies that we have cited in the revised or the previous manuscript have shown that exclusive breastfeeding is more effective than partial breastfeeding in reducing the maternal risks of metabolic syndrome or cardiovascular risk factors. The article suggested by Reviewer #3 examined the association between breastfeeding history and cardiovascular risk factors in children (Int J Obes (Lond). 2019;43(8):1568-1577.), and thus, we have not cited this article in the revised manuscript.

(4) Line 214: Why is this test not a linear regression? Didn't outcome and exposure have a normal distribution?

Response: Quantile-quantile plots indicated deviation from normality in all the continuous variables. Therefore, we conducted the Kruskal-Wallis test rather than linear regression to examine the associations between the total breastfeeding duration and the continuous variables (lines 219–222).

(5) Line 226: Even if it was the median age of the study, it does not characterize a strong justification regarding the stratification of the sample. I strongly suggest revising. Maybe it would be better to use more usual maternal age stratifications, separating younger and older mothers.

Response: We have changed the cut-off of the participants’ age from 57 to 55 years old as a usual maternal age in the revised manuscript (lines 235–239). The results of the reanalysis did not materially change from those of the previous manuscript.

(6) Line 235: This categorization of breastfeeding weeks is too confusing even to refer to the results of the study.

Response: As mentioned in the response to Reviewers #2’s comment (2), we have changed our breastfeeding duration measured in weeks to that measured in months. Additionally, we have again stated the categories of total breastfeeding duration at the beginning of the Results (lines 282–284).

(7) Table 1. It was not mentioned in the methods that there would be a description of the sample according to the breastfeeding categories. Indicate at the bottom of the table which test is used for p-value.

Response: In the previous and the revised manuscript (lines 220–223), we have written that associations between participant characteristics and the total breastfeeding duration were examined using the Kruskal-Wallis test for continuous variables and the chi-square test of independence for categorical variables. In accordance with this comment, we have added text to indicate that participants’ characteristics were expressed according to the categories of total breastfeeding duration (lines 217–219). We have also added the types of statistical tests to Table 1 in the revised manuscript.

(8) Table 2. I confess that this variable (Total breastfeeding duration weeks…) is still confused. The authors have enough information, this variable hinders more than helps in the interpretation of results.

Response: Our questionnaire asked about the longest breastfeeding duration for one child among all children and the number of breastfed children, but did not ask about breastfeeding duration for each child. The longest breastfeeding duration ignores breastfeeding durations that mothers could have experienced when they raised other children. Moreover, the number of breastfed children ignores all breastfeeding durations that mothers experienced. To overcome these limitations, we have introduced the variable “total breastfeeding duration.” As pointed out in comment (7) of Reviewer #2, the total breastfeeding duration also has a potential limitation due to misclassification, and thus, we have added this point to the Discussion of the revised manuscript (lines 498–501).

(9) Line 417: "Finally, the cross-sectional design of our study prevented exclusion of recall bias and reverse causality". Why? Need to justify this claim. 

Response: We have added the potential mechanism causing recall bias in the revised manuscript (lines 508–511). The mechanisms of reverse causality are described in the previous and the revised manuscript (lines 501–503). Regarding reverse causality, previous meta-analyses of observational studies showed that mothers with higher body mass index before pregnancy are less likely to initiate or maintain exclusive breastfeeding. In the present study, however, the body mass index at about 20 years old was rather positively associated with all the indices of breastfeeding history (lines 399–402).

(10) The authors must present a figure with the study sample flowchart, thus facilitating the understanding of the entire sample selection.

Response: We have added Figure 1, which is a study sample flowchart, to the revised manuscript.

(11) The number of references is extravagant! I suggest reviewing and reducing.

Response: We have reduced the number of references in the revised manuscript from 80 to 53.

(12) My concern is that they are adjusting for mediator in the wrong way. In addition, you need to check the need for all the variables presented.

Response: We have selected our covariates based on previous studies that reported the associations between these variables and the risk of metabolic syndrome. However, one cohort study reported the association between breastfeeding and early natural menopause (JAMA Network Open. 2020;3(1):e1919615), and a meta-analysis of observational studies and randomized controlled trials showed that menopause can increase the risks of metabolic syndrome and its components (Climacteric. 2017;20(6):583-591.) Thus, we have excluded menopausal status from the covariates as a possible mediator. We thought that the other covariates did not mediate the effect of breastfeeding history on the risk of metabolic syndrome and thus have retained these variables. This is because one of them occurred before delivery (i.e., educational attainment) or because we do not have evidence that breastfeeding history affects the remaining factors long after delivery (i.e., age, residential area, parity, smoking status, alcohol consumption, physical activity, and dietary habit, age at first delivery, and history of hormone replacement therapy).

(13) Authors need to consider the idea of creating a variable of exclusive breastfeeding and total breastfeeding, there are differences.

Response: Regretfully, as stated in the previous and the revised manuscript (lines 161–163), our questionnaire did not have items concerning exclusive breastfeeding, which has prevented examination of this variable.

(14) To justify the stratifications performed (maternal age, for example) the authors need to test and present the findings for interaction.

Response: Concerning the associations between breastfeeding history and metabolic syndrome, which was our primary outcome, we have conducted likelihood ratio tests for the product terms of breastfeeding history and age, and all the results were statistically significant (lines 239–242 and 336–340).

Finally, we must report that the numbers of participants classified as missing in several covariates (alcohol consumption, daily total and leisure-time physical activity, and the modified Japanese Diet Index) (Table 1) and those having dyslipidemia (Table 3) have decreased in the revised manuscript. The former occurred because of the incorrect treatment of missing values, and the latter occurred due to a coding error defining dyslipidemia. When we calculate alcohol consumption, daily total and leisure-time physical activity, and the modified Japanese Diet Index, missing values in some items should be treated as zero based on the rules of questionnaires.

Again, we would like to thank the editors and the reviewers for their valuable comments. We hope that the revised manuscript is now suitable for publication in PLOS ONE.

Sincerely,

Takashi Matsunaga, MMS, RPT

Department of Preventive Medicine,

Nagoya University Graduate School of Medicine

---

## [Decision Letter · Decision Letter 1]

21 Jul 2021

PONE-D-21-02465R1

Associations of breastfeeding history with metabolic syndrome and cardiovascular risk factors in community-dwelling parous women: The Japan Multi-Institutional Collaborative Cohort Study

PLOS ONE

Dear Dr. Matsunaga,

Thank you for submitting your manuscript to PLOS ONE. After careful consideration, we feel that it has merit but does not fully meet PLOS ONE’s publication criteria as it currently stands. Therefore, we invite you to submit a revised version of the manuscript that addresses the points raised during the review process.

We look forward to receiving your revised manuscript.

Kind regards,

Antonio Palazón-Bru, PhD

Academic Editor

PLOS ONE

Journal Requirements:

Additional Editor Comments (if provided):

Reviewers' comments:

Reviewer's Responses to Questions

**Comments to the Author**

1. If the authors have adequately addressed your comments raised in a previous round of review and you feel that this manuscript is now acceptable for publication, you may indicate that here to bypass the “Comments to the Author” section, enter your conflict of interest statement in the “Confidential to Editor” section, and submit your "Accept" recommendation.

Reviewer #2: (No Response)

Reviewer #3: All comments have been addressed

2. Is the manuscript technically sound, and do the data support the conclusions?

Reviewer #2: Yes

Reviewer #3: Partly

3. Has the statistical analysis been performed appropriately and rigorously? 

Reviewer #2: Yes

Reviewer #3: No

4. Have the authors made all data underlying the findings in their manuscript fully available?

Reviewer #2: Yes

Reviewer #3: Yes

5. Is the manuscript presented in an intelligible fashion and written in standard English?

Reviewer #2: Yes

Reviewer #3: Yes

6. Review Comments to the Author

Reviewer #2: OVERALL COMMENTS TO THE AUTHORS

Thank you for your responses to comments received. Overall, the manuscript has improved as a result of changes made. I had a few more additional specific comments for the authors to consider.

SPECIFIC COMMENTS TO THE AUTHORS

Abstract

Line 66 Cardiovascular risk factors could be listed for added clarity

Results Suggest adding a sentence about findings in women >55 years.

Methods

Lines 147-148 Please provide a reference

Lines 191-200 How were the categories for alcohol intake and physical activity derived, and for women <55 and ≥55 years? Please provide any relevant references.

Results

Tables could not be seen in their entirety (perhaps a layout issue) which made review difficult.

Discussion

- While the authors have made an effort to make additions to the discussion based on comments received, these additions could be written more succinctly, especially lines 471-484.

- As previously raised, the total breastfeeding duration variable is limited as it is based on the product of the number of breastfed children and the longest breastfeeding duration in one child, and this variable may not accurately reflect the overall breastfeeding duration across children as breastfeeding duration may vary across children from the same mother. Is there any literature about how breastfeeding duration may vary among children from the same mother, in Japan or other similar countries? Please discuss the limitation of the total breastfeeding duration variable in light of any additional insights provided by relevant literature.

Reviewer #3: The manuscript looks interesting, but needs extensive methodological review.

The issues about breastfeeding are fragile, in addition to not having information on exclusive breastfeeding that is strongly associated with cardiometabolic outcomes

Reference: Victora CG, Bahl R, Barros AJ, França GV, Horton S, Krasevec J, Murch S, Sankar MJ, Walker N, Rollins NC; Lancet Breastfeeding Series Group. Breastfeeding in the 21st century: epidemiology, mechanisms, and lifelong effect. Lancet. 2016 Jan 30;387(10017):475-90. doi: 10.1016/S0140-6736(15)01024-7. PMID: 26869575.

I suggest that the authors work with the continuous breastfeeding duration variable as well and not in tertiles

Tables are cut, it was not possible to perform a full evaluation

Review analytics to avoid adjustments for association mediators

Regarding the temporality of events, one should consider the possibility that some of the evaluated outcomes precede the breastfeeding behavior

Study limitations should be described and discussed only in the discussion section.

7. PLOS authors have the option to publish the peer review history of their article (what does this mean?). If published, this will include your full peer review and any attached files.

Reviewer #2: No

Reviewer #3: No

---

## [Author Response · Author response to Decision Letter 1]

28 Aug 2021

August 28, 2021

Antonio Palazón-Bru, Ph.D.

Academic Editor

PLOS ONE

Dear Dr Palazón-Bru:

Responses to the reviewers’ comments

To facilitate your review of our revisions, the following are our point-by-point responses to the questions and comments provided in your e-mail dated July 23, 2021. Amendments that we have made are shown in colored text throughout the Revised Article with Track Changes.

Reviewer #2’s comments

Abstract

(1) Line 66 Cardiovascular risk factors could be listed for added clarity.

Response: We have added lists of the cardiovascular risk factors (Lines 66–67 in the unmarked manuscript).

(2) Results Suggest adding a sentence about findings in women >55 years.

Response: Following this suggestion, we have added a sentence about the results in women >55 years of age (Lines 80–82).

Methods

(3) Lines 147-148 Please provide a reference.

Response: We have included a relevant reference of a systematic review (Line 150–152).

(4) Lines 191-200 How were the categories for alcohol intake and physical activity derived, and for women <55 and ≥55 years? Please provide any relevant references.

Response: According to this recommendation, we have added explanations that these covariates had been categorized based on the distributions of the present population and the cut-offs generally agreed with those reported by previous Japanese studies (Lines 210–212).

Results

(5) Tables could not be seen in their entirety (perhaps a layout issue) which made review difficult.

Response: In the revised article, we have embedded the tables to fit in one page, but this made the fonts rather small. Thus, we also have submitted the tables as supplementary materials (S Tables 1–3.xlsx). If the reviewers cannot read the fonts of the tables in the main text, please see the supplementary materials.

Discussion

(6) While the authors have made an effort to make additions to the discussion based on comments received, these additions could be written more succinctly, especially lines 471-484.

Response: We assume that reviewer #2 means the part concerning biological mechanisms behind the inverse associations of breastfeeding with metabolic syndrome and cardiovascular risk factors and have written this part as succinctly as possible (Lines 491–505).

(7) As previously raised, the total breastfeeding duration variable is limited as it is based on the product of the number of breastfed children and the longest breastfeeding duration in one child, and this variable may not accurately reflect the overall breastfeeding duration across children as breastfeeding duration may vary across children from the same mother. Is there any literature about how breastfeeding duration may vary among children from the same mother, in Japan or other similar countries? Please discuss the limitation of the total breastfeeding duration variable in light of any additional insights provided by relevant literature.

Response: We have included the results of a national survey in Japan and its limitations. Moreover, we have mentioned that the total breastfeeding duration of the present study probably had overestimated the true duration, and that the direction of the bias derived from this overestimation is difficult to ascertain. (Lines 516–521).

Reviewer #3’s comments

(1) The issues about breastfeeding are fragile, in addition to not having information on exclusive breastfeeding that is strongly associated with cardiometabolic outcomes.

Reference: Victora CG, Bahl R, Barros AJ, França GV, Horton S, Krasevec J, Murch S, Sankar MJ, Walker N, Rollins NC; Lancet Breastfeeding Series Group. Breastfeeding in the 21st century: epidemiology, mechanisms, and lifelong effect. Lancet. 2016 Jan 30;387(10017):475-90. doi: 10.1016/S0140-6736(15)01024-7. PMID: 26869575.

Response: The article suggested by reviewer #3 states that a long breastfeeding duration can lead to lower risks of breast and ovarian cancers, osteoporosis, and type 2 diabetes mellitus in mothers, but it does not compare the effects of exclusive breastfeeding to those of partial breastfeeding in reducing the risks of these diseases (“Effects on women who breastfed” part on the table and pages 484–485 in the suggested article). We have cited this article in the Discussion, but have stated that few previous studies examined this issue related to the maternal risks of metabolic syndrome and cardiovascular risk factors (Lines 521–527).

(2) I suggest that the authors work with the continuous breastfeeding duration variable as well and not in tertiles.

Response: Following this suggestion, we conducted a sensitivity analysis that examined the associations between each measurement of breastfeeding history and metabolic syndrome by incorporating these measurements into models as continuous variables. The results remained almost the same as those of the trend tests that incorporated ordinal variables measuring breastfeeding duration using tertiles (Lines 250–254, 392–395, and S1 Table). We have also mentioned that the measurements of breastfeeding history were categorized to deal with potential nonlinear associations in the Exposure part of the Methods (Lines 166–170).

(3) Tables are cut, it was not possible to perform a full evaluation.

Response: In the revised article, we have embedded the tables to fit in one page, but this made the fonts rather small. Thus, we also have submitted the tables as supplementary materials (S Tables 1–3.xlsx). If the reviewers cannot read the fonts of the tables in the main text, please see the supplementary materials.

(4) Review analytics to avoid adjustments for association mediators.

Response: We agree that several confounding factors in our study (smoking status, alcohol consumption, daily total physical activity, leisure-time physical activity, the modified Japanese Diet Index, and history of hormone replacement therapy) probably followed breastfeeding history in time and could be association mediators. Thus, we conducted a sensitivity analysis that examined associations between each indicator of breastfeeding history and metabolic syndrome by incorporating only covariates that had preceded breastfeeding history in time (Lines 254–257). The results were generally unchanged from those in the multivariable-adjusted model of Table 2 (Lines 395–399 and S2 Table). We also have added a discussion about a bias developed from adjusting for potential mediators (Lines 540–548). Finally, in the Discussion, we have mentioned the need for prospective assessment of potential confounding factors to obviate this bias (Lines 549–551).

(5) Regarding the temporality of events, one should consider the possibility that some of the evaluated outcomes precede the breastfeeding behavior.

Response: In response to this comment, we added a discussion about another possible example of reverse causality other than that due to obesity before pregnancy, where gestational diabetes may prevent the initiation and maintenance of breastfeeding (Lines 531–534).

(6) Study limitations should be described and discussed only in the discussion section.

Response: We assume that reviewer #3 is referring to the part about no data on exclusive breastfeeding and the approximation of the total breastfeeding duration stated in the Methods. We have added this limitation in the Discussion (Lines 512–527). For clarity, however, we also retained the description in the Exposure part of the Methods (Lines 162–166).

Again, we would like to thank the editors and the reviewers for their valuable comments. We hope that the revised manuscript is now suitable for publication in PLOS ONE.

Sincerely,

Takashi Matsunaga, MMS, RPT

Department of Preventive Medicine,

Nagoya University Graduate School of Medicine

---

## [Decision Letter · Decision Letter 2]

30 Sep 2021

PONE-D-21-02465R2Associations of breastfeeding history with metabolic syndrome and cardiovascular risk factors in community-dwelling parous women: The Japan Multi-Institutional Collaborative Cohort StudyPLOS ONE

Dear Dr. Matsunaga,

Thank you for submitting your manuscript to PLOS ONE. After careful consideration, we feel that it has merit but does not fully meet PLOS ONE’s publication criteria as it currently stands. Therefore, we invite you to submit a revised version of the manuscript that addresses the points raised during the review process.

We look forward to receiving your revised manuscript.

Kind regards,

Antonio Palazón-Bru, PhD

Academic Editor

PLOS ONE

Reviewers' comments:

Reviewer's Responses to Questions

**Comments to the Author**

1. If the authors have adequately addressed your comments raised in a previous round of review and you feel that this manuscript is now acceptable for publication, you may indicate that here to bypass the “Comments to the Author” section, enter your conflict of interest statement in the “Confidential to Editor” section, and submit your "Accept" recommendation.

Reviewer #2: All comments have been addressed

Reviewer #3: All comments have been addressed

2. Is the manuscript technically sound, and do the data support the conclusions?

Reviewer #2: Yes

Reviewer #3: Partly

3. Has the statistical analysis been performed appropriately and rigorously? 

Reviewer #2: Yes

Reviewer #3: No

4. Have the authors made all data underlying the findings in their manuscript fully available?

Reviewer #2: No

Reviewer #3: No

5. Is the manuscript presented in an intelligible fashion and written in standard English?

Reviewer #2: (No Response)

Reviewer #3: Yes

6. Review Comments to the Author

Reviewer #2: Overall comments to the authors

Previous comments have been addressed. I only had one minor comment to add.

Discussion

Line 443 The sentence states that "although no systematic review or meta-analysis has been conducted". However, I am aware of a systematic review that examines breastfeeding and maternal cardiovascular risk factors and cardiovascular disease, and that mentions effects based on age. Breastfeeding and maternal cardiovascular risk factors and outcomes: A systematic review. PLoS One. 2017; 12(11): e0187923. The authors could check for other systematic reviews/meta-analyses, just in case.

Reviewer #3: (2) I suggest that the authors work with the continuous breastfeeding duration variable as well and

not in tertiles.

It would be interesting to present these sensitivity analyses.

(4) Review analytics to avoid adjustments for association mediators.

It would be interesting to present these sensitivity analyses.

Adjustment for mediators is considered a serious error because such variables are in the middle of the causal chain between exposure and outcome. Considering this, if the authors decide to keep these adjustments, my suggestion is that a robust mediation analysis be performed.

7. PLOS authors have the option to publish the peer review history of their article (what does this mean?). If published, this will include your full peer review and any attached files.

Reviewer #2: No

Reviewer #3: No

---

## [Author Response · Author response to Decision Letter 2]

29 Oct 2021

October 29, 2021

Antonio Palazón-Bru, Ph.D.

Academic Editor

PLOS ONE

Dear Dr Palazón-Bru:

Thank you for your e-mail dated September 30, 2021. We appreciated the reviewers’ thoughtful comments. To facilitate your review of our revisions, the following are our point-by-point responses to the questions and comments provided in your e-mail. Amendments that we have made are shown in colored text throughout the “Revised Article with Track Changes. docx.” file.

Responses to the reviewers’ comments

Reviewer #2’s comments

Discussion

Line 443 The sentence states that "although no systematic review or meta-analysis has been conducted". However, I am aware of a systematic review that examines breastfeeding and maternal cardiovascular risk factors and cardiovascular disease, and that mentions effects based on age. Breastfeeding and maternal cardiovascular risk factors and outcomes: A systematic review. PLoS One. 2017; 12(11): e0187923. The authors could check for other systematic reviews/meta-analyses, just in case.

Response: We apologize for having overlooked the suggested article and cited this article in the Discussion in the revised manuscript (ref. 37, lines 446–449). Another systematic review investigated whether the duration between delivery and the study or length of follow-up influences the association between breastfeeding and hypertension with a cut-off of 6 months (Bonifacino E et al. Breastfeed Med. 2018;13(9):578-588.) However, this cut-off is too short to examine the effects of breastfeeding on metabolic syndrome in middle-aged or older women, and thus we did not cite this systematic review.

Reviewer #3’s comments

 If the tables cannot be seen in the main text, please refer to the Excel files uploaded to the submission and review system.

(2) I suggest that the authors work with the continuous breastfeeding duration variable as well and

not in tertiles. It would be interesting to present these sensitivity analyses.

Response: In the previous and the revised manuscript, we have conducted the suggested sensitivity analysis (S1 Table). In both the younger and the older age groups, the results remained almost the same as those of the trend tests in the main analysis (Table 2). The number of breastfed children was inversely associated with metabolic syndrome in the younger age group, whereas the longest and total breastfeeding duration was positively correlated with the syndrome in the older age group. In all participants, the longest and total breastfeeding duration was also correlated with the syndrome in this sensitivity analysis, which was different from results of the main analysis with the tertiles (S1 Table and lines 250–254 and 392–400).

(4) Review analytics to avoid adjustments for association mediators. It would be interesting to present these sensitivity analyses. Adjustment for mediators is considered a serious error because such variables are in the middle of the causal chain between exposure and outcome. Considering this, if the authors decide to keep these adjustments, my suggestion is that a robust mediation analysis be performed.

Response: As a sensitivity analysis, we excluded potential mediators from the covariates (S2 Table). The directions of associations were unchanged from those of the main analysis as we presented the result in the previous and the present manuscript (Table 2).

 Regarding mediation analysis, however, we disagree with reviewer #3’s suggestion for three reasons. First, adjustments for descendants of exposure induce biases, but those for factors in the future of exposure could reduce biases (Hernán MA, Robins JM. Causal inference: what if. pp95. Available from: https://cdn1.sph.harvard.edu/wp-content/uploads/sites/1268/2021/03/

ciwhatif_hernanrobins_30mar21.pdf). As a simulation study has shown, if the covariates of the present study are affected by unmeasured confounding variables and the effects of confounding are strong, not adjusting for these covariates may lead to residual confounding even if these covariates are also mediators. This simulation study recommends presenting both results adjusted and unadjusted for potential mediators that are also descendants of confounding variables when only variables that concede exposure were measured and sufficient knowledge is not available about relationships between variables examined in studies (Groenwold RHH et al. Epidemiology. 2021;32(2):194-201.) Following this recommendation, we have retained all the covariates in main models and conducted the sensitivity analysis mentioned above (lines 554–562). The directions of the results were unchanged when we excluded potential mediators from the covariates (S2 Table and lines 400–404). Second, the present study aimed to investigate neither the direct nor indirect effects but the total effect of breastfeeding on metabolic syndrome. The investigation of total effects does not require mediation analysis. Even if the covariates of the present study are potent mediators, the additional analysis provides total effect estimates that avoid the bias from the adjustments for these covariates (S2 Table). Third, our study measured few potential confounding factors that preceded breastfeeding history. Even if we conduct mediation analysis, we should not obtain valid estimates of the mediating effects without sufficient adjustments for confounding factors. Thus, we did not conduct a mediation analysis.

Again, we would like to thank the editors and the reviewers for their valuable comments. We hope that the revised manuscript is now suitable for publication in PLOS ONE.

Sincerely,

Takashi Matsunaga, MMS, RPT

Department of Preventive Medicine,

Nagoya University Graduate School of Medicine

---

## [Editor Report · Decision Letter 3]

21 Dec 2021

Associations of breastfeeding history with metabolic syndrome and cardiovascular risk factors in community-dwelling parous women: The Japan Multi-Institutional Collaborative Cohort Study

PONE-D-21-02465R3

Dear Dr. Matsunaga,

We’re pleased to inform you that your manuscript has been judged scientifically suitable for publication and will be formally accepted for publication once it meets all outstanding technical requirements.

Kind regards,

Antonio Palazón-Bru, PhD

Academic Editor

PLOS ONE

Additional Editor Comments (optional):

I have checked your responses and from my point of view, your work has good quality to be accepted in its current form in PLoS One. Congratulations!
---

## [Editor Report · Acceptance letter]

27 Dec 2021

PONE-D-21-02465R3 

Associations of breastfeeding history with metabolic syndrome and cardiovascular risk factors in community-dwelling parous women: The Japan Multi-Institutional Collaborative Cohort Study 

Dear Dr. Matsunaga:

I'm pleased to inform you that your manuscript has been deemed suitable for publication in PLOS ONE. Congratulations! Your manuscript is now with our production department. 

Kind regards, 

on behalf of

Dr. Antonio Palazón-Bru 

Academic Editor

PLOS ONE